# Short-term exposure to wildfire-related PM$_{2.5}$ increases mortality risks and burdens in Brazil

Tingting Ye [1], Rongbin Xu [1], Xu Yue [2], Gongbo Chen[1], Pei Yu [1], Micheline S. Z. S. Coêlho [3], Paulo H. N. Saldiva[3], Michael J. Abramson [1], Yuming Guo [1] ✉ & Shanshan Li[1] ✉

To assess mortality risks and burdens associated with short-term exposure to wildfire-related fine particulate matter with diameter ≤ 2.5 μm (PM$_{2.5}$), we collect daily mortality data from 2000 to 2016 for 510 immediate regions in Brazil, the most wildfire-prone area. We integrate data from multiple sources with a chemical transport model at the global scale to isolate daily concentrations of wildfire-related PM$_{2.5}$ at a 0.25 × 0.25 resolution. With a two-stage time-series approach, we estimate (i) an increase of 3.1% (95% confidence interval [CI]: 2.4, 3.9%) in all-cause mortality, 2.6% (95%CI: 1.5, 3.8%) in cardiovascular mortality, and 7.7% (95%CI: 5.9, 9.5) in respiratory mortality over 0–14 days with each 10 μg/m³ increase in daily wildfire-related PM$_{2.5}$; (ii) 0.65% of all-cause, 0.56% of cardiovascular, and 1.60% of respiratory mortality attributable to acute exposure to wildfire-related PM$_{2.5}$, corresponding to 121,351 all-cause deaths, 29,510 cardiovascular deaths, and 31,287 respiratory deaths during the study period. In this study, we find stronger associations in females and adults aged ≥ 60 years, and geographic difference in the mortality risks and burdens.

Wildfires have become an increasingly visible and potent threat in Brazil, modulated by both climate and human-driven land use changes[1]. People deliberately set fires to clear forest and other vegetation to prepare and maintain land for agriculture[2]. In Amazon, fire emissions are greater in years with higher deforestation rates. A number of severe drought episodes have occurred during 2005, 2007, and 2010, and fire emissions increased 1.5–2.8 fold compared with non-drought years[3]. Biomass burning is the dominant source of particulate matter over the Amazon[4]. During the dry season (August to October) when there are a large number of vegetation fires, regional mean concentrations of fine particulate matter with diameters ≤2.5 μm (PM$_{2.5}$) in heavily impacted sites in south-western Amazon can exceed 33 μg/m[35] with daily mean peak concentrations close to 100 μg/m[36]. In contrast, during wet seasons when there is little fire activity, PM$_{2.5}$ concentrations in south-western Amazon were one order of magnitude lower, and could be as low as 1.5 μg/m³ across the central Amazon[5]. In the western Amazon region, Butt and colleagues estimated that vegetation fires contributed 80% of mean surface PM$_{2.5}$ during the dry season[7]. Smoke produced during such episodes extends wildfire impacts far beyond the vicinity of the flames in the form of air pollution, of which particulate matter spreads much farther[8].

Exposure to wildfire-related PM$_{2.5}$ has been shown to have adverse impacts on human health[9]. Our previous work on wildfire-related PM$_{2.5}$ and daily mortality in 749 cities showed that all-cause

[1]Climate, Air Quality Research Unit, School of Public Health and Preventive Medicine, Monash University, Melbourne, VIC 3004, Australia. [2]Jiangsu Key Laboratory of Atmospheric Environment Monitoring and Pollution Control, Collaborative Innovation Center of Atmospheric Environment and Equipment Technology, School of Environmental Science and Engineering, Nanjing University of Information Science & Technology (NUIST), Nanjing 210044, China. [3]Urban Health Laboratory University of São Paulo, Faculty of Medicine/INSPER, São Paulo 01246-903, Brazil. ✉e-mail: yuming.guo@monash.edu; shanshan.li@monash.edu

mortality increased by 1.9% (95%CI 1.6–2.2), cardiovascular mortality by 1.7% (1.2–2.1), and respiratory mortality by 1.9% (1.3–2.5) with every 10 μg/m³ increase in PM$_{2.5}$ at lag 0–2 days[10]. The stronger effect of wildfire-related PM$_{2.5}$ than PM$_{2.5}$ from other sources (e.g., urban sources) has been observed in recent epidemiological studies[11,12]. For example, Aguilera et al. isolated wildfire-specific PM$_{2.5}$ using a series of statistical approaches and exposure definitions, and found higher increases in respiratory hospitalizations with increase in wildfire-specific PM$_{2.5}$ compared to the associations with non-wildfire PM$_{2.5}$[11]. The higher toxicity of biomass particles on children's respiratory health was reported in a recent study[13]. Toxicological studies have shown differences in the toxicological characteristics of aerosols from different sources[14,15], and biomass particles exhibits greater toxicity in comparison with those produced by fossil fuels.

Some studies from the past 5 years in Brazil have attempted to assess the premature deaths attributable to air pollution from fire emissions[3,7,16,17]. However, they only used total particulate matter concentrations during wildfire events and the risk of death from all-cause or non-accidental deaths. The association between wildfire-specific PM$_{2.5}$ and the risk of deaths from specific causes (e.g., respiratory, cardiovascular causes) remained uncertain. Existing evidence was limited to single-city or single-region near the location of fire sources during burning seasons due to known consequences of biomass burning and emissions. Little was known about the magnitude and scope of its adverse effects on air quality and public health across the nation and whether certain populations were more susceptible. A comprehensive understanding of health effects of source-specific PM$_{2.5}$, such as from wildfires, would inform policy.

To address these research gaps, this study used a nationwide mortality data set and aimed to quantify the cause-specific, demo-graphic, and temporal variations in the associations between daily exposure to wildfire-related PM$_{2.5}$ and risk of mortality in Brazil.

## Results

Table 1 summarizes daily mean temperature, relative humidity, wildfire-related PM$_{2.5}$, and cause-specific deaths for 510 immediate regions in Brazil. During the study period, there were 18,681,906 all-cause deaths, among which 5,271,936 were cardiovascular and 1,954,849 respiratory. Mean daily counts of all-cause, cardiovascular and respiratory deaths were 3008, 849, and 315, respectively (Table 1). Adults aged ≥60 years accounted for 61.4% of the total all-cause mortality, and females for 42.7%. Though the concentrations of wildfire-related PM$_{2.5}$ in the Brazil shift with seasons and regions (Fig. 1 and Supplementary Fig. 1), all months experienced wildfire smoke. The average daily mean wildfire-related PM$_{2.5}$ of the 510 immediate regions was 2.8 μg/m³ (standard deviation [SD] 2.7 μg/m³) between 2000 and 2016.

### Wildfire-related PM$_{2.5}$-mortality associations

The associations between wildfire-related PM$_{2.5}$ and cause-specific mortalities were linear (Supplementary Fig. 2). Figure 2 presents the pooled estimated effects for each lag of 0 to 14 days for all-cause, cardiovascular, and respiratory mortality associated with a 10 μg/m³ increase in wildfire-related PM$_{2.5}$. At the national level, the pooled effects of wildfire-related PM$_{2.5}$ exposure were acute, followed by temporal displacement on lag days 2–6 where some expected deaths might happen in advance on lag days 0–1 due to exposure on lag day 0. Thus, we used 14 days as maximum lag to fully capture the lag effects.

Cumulative RRs over lag 0–14 days for cause- and group- specific mortality are shown in Fig. 3. Wildfire-related PM$_{2.5}$ significantly increased the risks of all-cause, cardiovascular, and respiratory mortality, as well as all-cause mortality in different age and sex groups. The associations between increased mortality risks from respiratory diseases and short-term exposure to wildfire-related PM$_{2.5}$ were stronger. For the total population, every 10 μg/m³ increase in daily mean wildfire-related PM$_{2.5}$ was associated with 3.1 (95%CI: 2.4–3.9, $p < 0.001$) increase in all-cause mortality, 2.6 (1.5–3.8, $p < 0.001$) increase in cardiovascular mortality, and 7.7 (5.9–9.5, $p < 0.001$) for respiratory mortality. Age- and sex-stratified analysis revealed that adults aged 60 years or older and females appeared more sensitive to acute impacts of wildfire-related PM$_{2.5}$ exposure, and the differences were statistically significant for all-cause and respiratory related mortality ($p$-value for difference <0.05, Fig. 2). Geographic difference in the cumulative effect estimates were observed, ranging from the lowest in the North [−0.8 (−2.9–1.2)] to the highest in the Southeast [6.0 (4.7–7.2)] for all-cause mortality.

### Attributable health burdens

Attributable mortality fractions, rates and attributable deaths showed the mortality burdens based on the pooled national associations between mortality and wildfire-related PM$_{2.5}$ exposure over 0–14 lag days (Table 2). An estimated 130,273 all-cause deaths (95% CI: 76,534–183,346), 32961 cardiovascular deaths (7628–57,756), and 33,807 respiratory deaths (19,225–47,919) were attributable to acute wildfire-related PM$_{2.5}$ exposure, corresponding to fractions of 0.70% (0.41–0.98) for all-causes, 0.63% (0.14–1.10) for cardiovascular, and 1.73% (0.98–2.45) for respiratory mortality during the study period. Between sex groups, the estimated attributable rate was slightly higher in females in comparison to males from all-cause to cause-specific mortality (Table 2). Compared to people younger than 60 years, sub-stantially higher attributable mortality rates due to wildfire-related PM$_{2.5}$ were observed in those aged 60 years or older, corresponding to 383.9 all-cause deaths (215.4–549.7), 107.7 cardiovascular (13.1–199.8), and 117.4 respiratory deaths (67.2–165.8) per million residents per year. The estimated attributable mortality rate was the highest in the Central West and followed by the Southeast.

**Table 1 | Summary statistics of daily mean values of meteorological variables, air pollutants and cause-specific death counts in Brazil**

| Variables | Mean ± SD | Min | $P_{25}$ | $P_{50}$ | $P_{75}$ | Max |
|---|---|---|---|---|---|---|
| Temperature (°C) | 23.6 ± 1.8 | 16.7 | 22.3 | 24.0 | 25.0 | 27.7 |
| Relative humidity (%) | 74.1 ± 5.4 | 56.6 | 70.6 | 75.0 | 78.1 | 86.8 |
| Wildfire-related PM$_{2.5}$ (μg/m³) | 2.8 ± 2.7 | 0.2 | 1.0 | 1.7 | 3.5 | 19.8 |
| Causes and subgroups | | | | | | |
| All-causes | 3008 ± 342 | 2235 | 2733 | 2977 | 3252 | 4188 |
| Females | 1283 ± 167 | 919 | 1148 | 1266 | 1403 | 1898 |
| Males | 1723 ± 185 | 1266 | 1580 | 1709 | 1852 | 2413 |
| Age 0–59 years | 1139 ± 94 | 903 | 1072 | 1123 | 1197 | 1638 |
| Age ≥60 years | 1847 ± 308 | 1226 | 1595 | 1819 | 2066 | 2929 |
| Respiratory | 315 ± 71 | 162 | 260 | 304 | 357 | 587 |
| Females | 148 ± 38 | 65 | 118 | 142 | 171 | 299 |
| Males | 167 ± 35 | 91 | 141 | 163 | 188 | 311 |
| Age 0–59 years | 65 ± 12 | 33 | 56 | 63 | 71 | 141 |
| Age ≥60 years | 249 ± 66 | 110 | 198 | 239 | 290 | 502 |
| Cardiovascular | 849 ± 108 | 585 | 767 | 850 | 926 | 1,267 |
| Females | 403 ± 54 | 248 | 363 | 403 | 442 | 611 |
| Males | 446 ± 58 | 301 | 402 | 445 | 488 | 680 |
| Age 0–59 years | 187 ± 18 | 127 | 175 | 187 | 199 | 255 |
| Age ≥60 years | 659 ± 99 | 423 | 583 | 661 | 730 | 1032 |

*SD* standard deviation, $P_{25}$ the 25th percentile, $P_{50}$ the 50th percentile, $P_{75}$ the 75th percentile, *PM$_{2.5}$* wildfire-related fine particulate matter with diameter ≤2.5 μm.

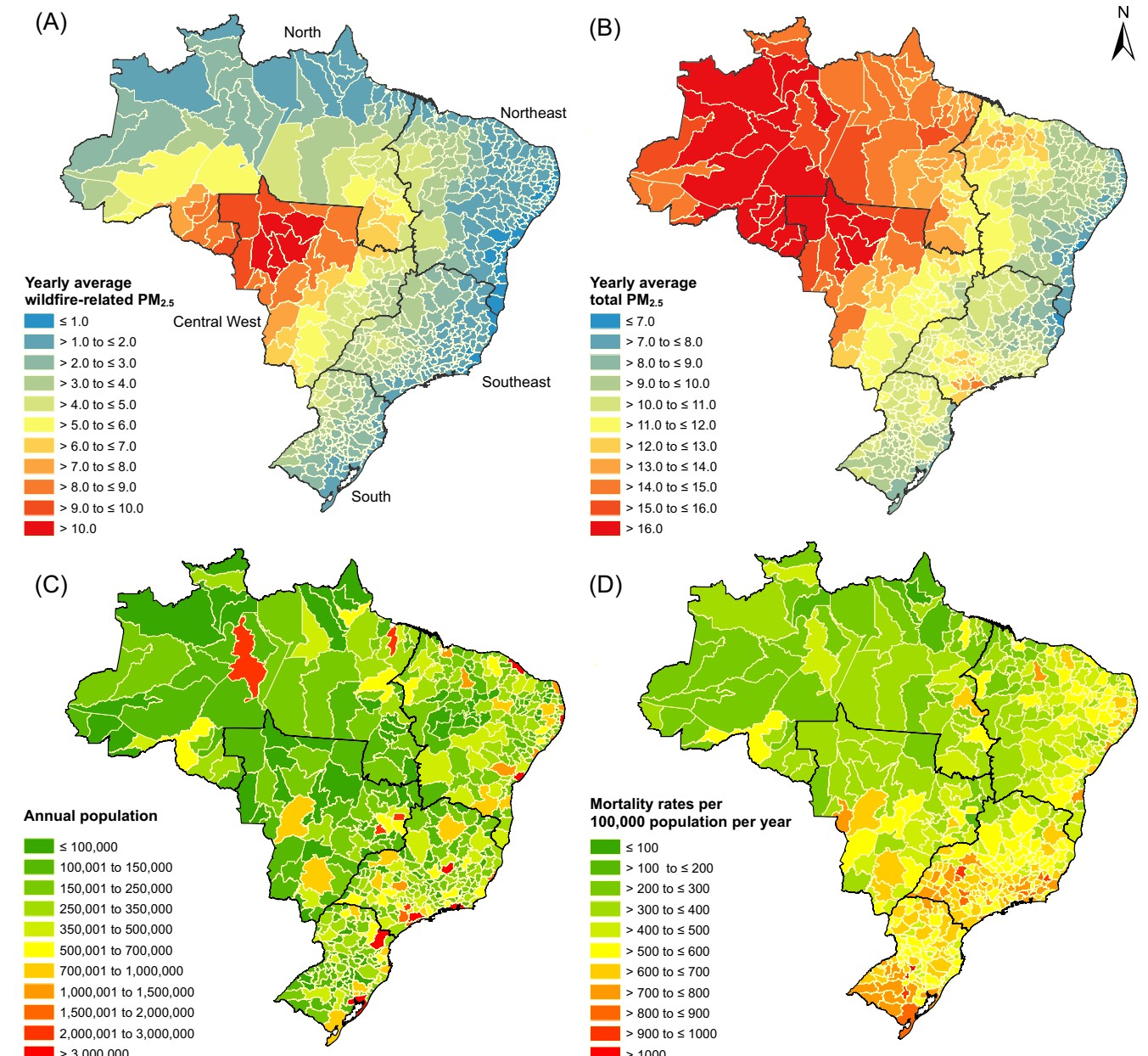

**Fig. 1 | Spatial variations in air pollutants, population, and mortality rates in Brazil.** There are regional differences in (**A**) yearly average wildfire-related PM$_{2.5}$ μg/m³, (**B**) yearly average total PM$_{2.5}$ μg/m³, (**C**) annual population, and (**D**) mortality rates/100,000 population/year across 510 immediate regions in Brazil during 2000–2016. (Note: PM$_{2.5}$, fine particulate matter with diameter ≤2.5 μm.).

## Result of sensitivity analyses

Sensitivity analyses showed that a maximum lag of 14 days was sufficient to capture the lagged effects of wildfire-related PM$_{2.5}$ (Supplementary Table 1). Our results were robust after changing the degrees of freedom of lag days in the cross-basis function, and the changes in the degrees of freedom for meteorological variables.

## Discussion

Using a large nationwide dataset, this study provides robust epidemiological evidence of the acute effects from wildfire-related PM$_{2.5}$ exposure on mortality. Assuming a causal exposure pathway, approximately 0.70% of Brazilian all-cause deaths, equivalent to 130,273 cases, could be attributable to wildfire-related PM$_{2.5}$ exposure during the study period, with an attributable mortality rate of 37.5 per million population. We found substantial adverse health effects both for cardiovascular and respiratory diseases, when estimates were pooled at a regional and national level, much like a previous study[18], with stronger effect estimates for respiratory mortality. We observed geographic variation in the relationship between wildfire-related PM$_{2.5}$ exposure and mortality, with individuals in the Southeast most susceptible, whereas those in the North and Northeast were less susceptible to wildfire-related PM$_{2.5}$. Similar result has been reported in our previous study for hospital admissions[19].

Females and those aged 60 years or older appeared to be more sensitive to wildfire-related PM$_{2.5}$ exposure in Brazil. The sex differences for all-cause and respiratory mortalities remain significant in ≥60 years age subgroups (Supplementary Fig. 3). A recent review and meta-analysis found a greater effect of wildfire-related PM$_{2.5}$ on respiratory health among females compared to males for asthma and chronic obstructive pulmonary disease[20]. There are also studies reporting higher risks for cardiovascular diseases in relation to wildfire smoke in females than males[21,22], but inconsistency remains[23]. Our previous

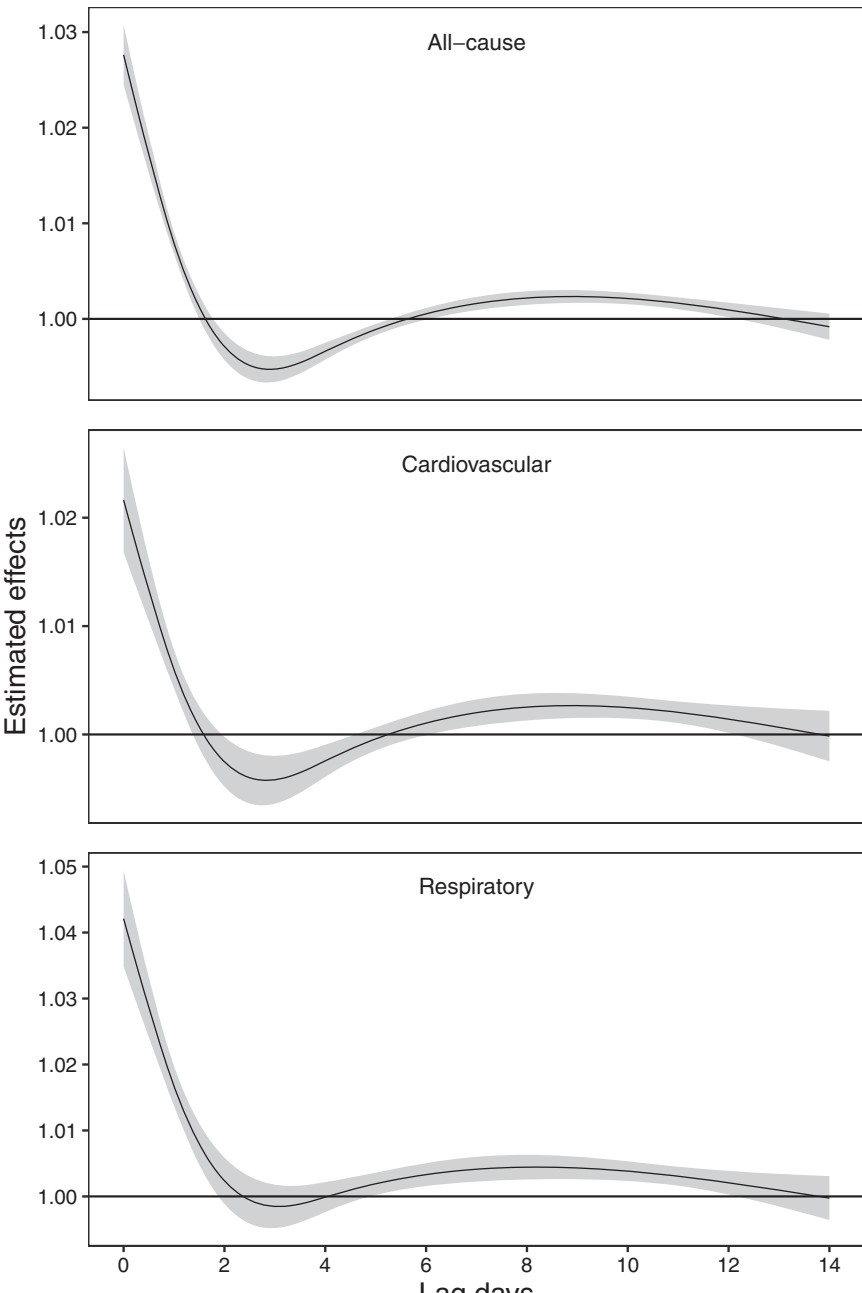

**Fig. 2 | The association between wildfire-related PM₂.₅ exposure (every 10 μg/ m³ increase in wildfire-related PM₂.₅) and all-cause, cardiovascular, and respiratory mortalities across 0–14 lag days.** (Note 1: The solid lines denote point-estimates and shaded areas denote the corresponding 95% confidence intervals. Note 2: estimates were derived from models with data in 510 immediate regions. Note 3: PM₂.₅, fine particulate matter with diameter ≤2.5 μm).

study found non-significant sex difference in hospital admissions for all-cause outcomes associated with short-term exposure to wildfire-related PM₂.₅[19]. In general, the findings for differential effects by sex are inconclusive and may vary by study period and population, exposure intensity and duration, and health outcome, which warrants further investigation. Many more studies agree with the differences by age groups, in that older adults are more vulnerable to ambient air pollution from wildfire smoke[24,25], with small variations of the age at which these effects are evident. Potential explanation could be the elderly have higher sensitivity and lower resistance to air pollution due to having more chronic diseases, and degraded immune systems.

The estimated attributed deaths of our study are consistent in magnitude with those of previous investigations for South America and Brazil, despite different effect estimates and exposure periods,

with strong evidence of acute adverse health outcomes due to exposure to wildfire-related PM₂.₅. We estimated that 130,273 deaths could be attributable to wildfire-related PM₂.₅ exposure from 2000 to 2016, equivalent to 7,663 deaths annually. Johnston et al estimated preventing fires would avoid 10,000 premature deaths annually between 1997 and 2006 in South America[26]. Reddington et al.[3] estimated prevention of vegetation fires would avert about 7000–9700 premature deaths annually across South America and 4200–5200 in Brazil between 2002 and 2011. A recent study estimated that vegetation fires in the Amazon basin in 2012, a year with emissions similar to the 11-year average (2008–2018), were linked to ~9770 premature deaths in Brazil[7]. Our estimations are robust and an advance due to the use of large nationwide time-series data, a regional model, and updated exposure-response relationship associated with wildfire-related PM₂.₅.

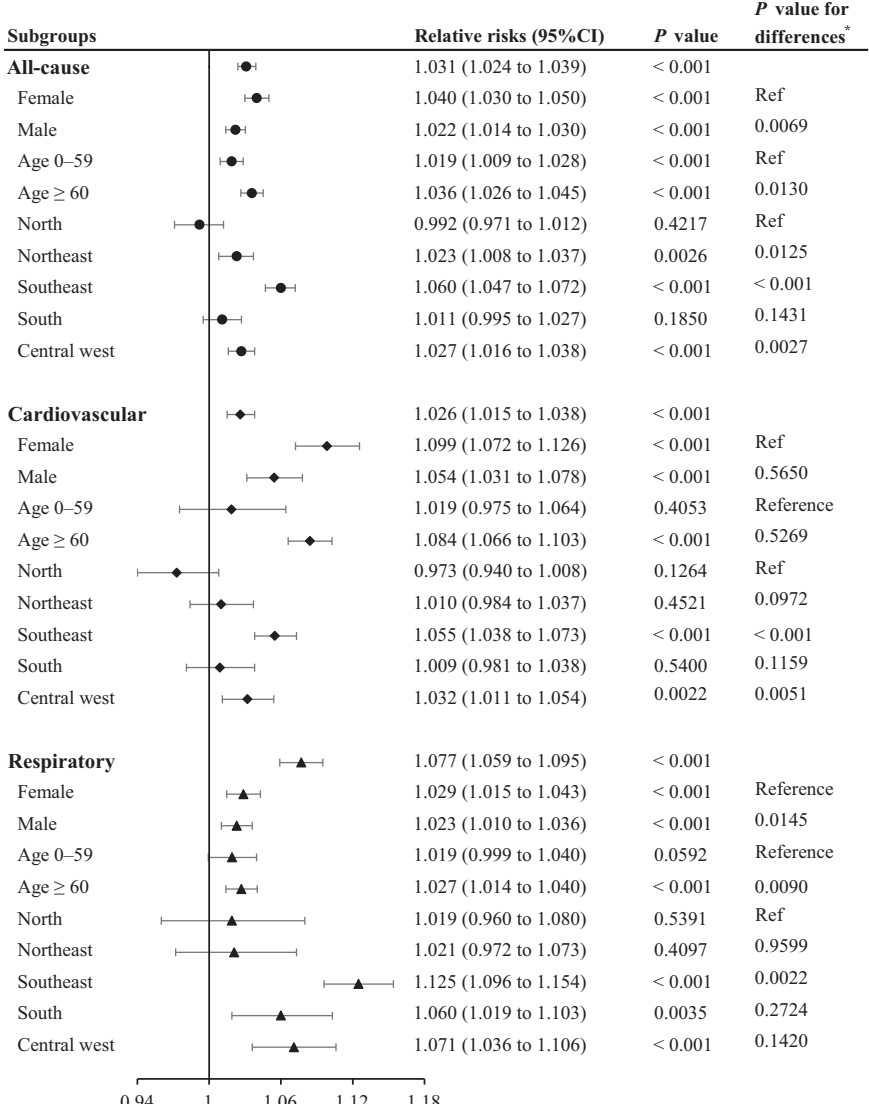

| Subgroups | Relative risks (95%CI) | P value | P value for differences* |
|---|---|---|---|
| **All-cause** | 1.031 (1.024 to 1.039) | < 0.001 | |
| Female | 1.040 (1.030 to 1.050) | < 0.001 | Ref |
| Male | 1.022 (1.014 to 1.030) | < 0.001 | 0.0069 |
| Age 0–59 | 1.019 (1.009 to 1.028) | < 0.001 | Ref |
| Age ≥ 60 | 1.036 (1.026 to 1.045) | < 0.001 | 0.0130 |
| North | 0.992 (0.971 to 1.012) | 0.4217 | Ref |
| Northeast | 1.023 (1.008 to 1.037) | 0.0026 | 0.0125 |
| Southeast | 1.060 (1.047 to 1.072) | < 0.001 | < 0.001 |
| South | 1.011 (0.995 to 1.027) | 0.1850 | 0.1431 |
| Central west | 1.027 (1.016 to 1.038) | < 0.001 | 0.0027 |
| **Cardiovascular** | 1.026 (1.015 to 1.038) | < 0.001 | |
| Female | 1.099 (1.072 to 1.126) | < 0.001 | Ref |
| Male | 1.054 (1.031 to 1.078) | < 0.001 | 0.5650 |
| Age 0–59 | 1.019 (0.975 to 1.064) | 0.4053 | Reference |
| Age ≥ 60 | 1.084 (1.066 to 1.103) | < 0.001 | 0.5269 |
| North | 0.973 (0.940 to 1.008) | 0.1264 | Ref |
| Northeast | 1.010 (0.984 to 1.037) | 0.4521 | 0.0972 |
| Southeast | 1.055 (1.038 to 1.073) | < 0.001 | < 0.001 |
| South | 1.009 (0.981 to 1.038) | 0.5400 | 0.1159 |
| Central west | 1.032 (1.011 to 1.054) | 0.0022 | 0.0051 |
| **Respiratory** | 1.077 (1.059 to 1.095) | < 0.001 | |
| Female | 1.029 (1.015 to 1.043) | < 0.001 | Reference |
| Male | 1.023 (1.010 to 1.036) | < 0.001 | 0.0145 |
| Age 0–59 | 1.019 (0.999 to 1.040) | 0.0592 | Reference |
| Age ≥ 60 | 1.027 (1.014 to 1.040) | < 0.001 | 0.0090 |
| North | 1.019 (0.960 to 1.080) | 0.5391 | Ref |
| Northeast | 1.021 (0.972 to 1.073) | 0.4097 | 0.9599 |
| Southeast | 1.125 (1.096 to 1.154) | < 0.001 | 0.0022 |
| South | 1.060 (1.019 to 1.103) | 0.0035 | 0.2724 |
| Central west | 1.071 (1.036 to 1.106) | < 0.001 | 0.1420 |

0.94   1   1.06   1.12   1.18

**Fig. 3 | Pooled relative risks of all-cause, cardiovascular, and respiratory mortality (stratified by sex and age groups) associated with a 10 µg/m³ increase in wildfire-related PM₂.₅ over lag 0–14 days.** (Note 1: *p value for the differences in cumulative relative risks (with 95% CI) across population subgroups were estimated by fixed effect meta-regression. Note 2: black error bars correspond to 95% confidence intervals, center for the error bars correspond to points estimate of RRs. Note 3: cause-, age- and sex-specific estimates were derived from the main model with data in 510 immediate regions. Estimates were pooled for each region, i.e., n = 62 in North, n = 154 in Northeast, n = 53 in Central West, n = 145 in Southeast, and n = 96 in South. Note 4: PM₂.₅, fine particulate matter with diameter ≤2.5 µm.).

These findings could aid in arousing awareness of the wildfire smoke crisis, and responses to protect health against wildfire-related air pollution. We suggest public agencies that are responsible for releasing advice regarding health protection against wildfire smoke educating residents keep track of air quality during fire season[27]. It is vital for residents living in areas potentially affected by wildfires to adjust their activities, and gather emergency supplies (e.g., foods, water, first aid medication) before a fire occurs. Personal protections are recommended, including wearing facemasks, avoiding heavy and prolonged outdoor activities, staying indoors, and keeping the windows closed, using air purifiers, especially for sensitive populations. For residents in the Amazon, a staying indoor strategy might be less effective or even impractical. We suggested further assistance from local government.

This study provides robust epidemiological evidence for mortality risk attributable to short-term exposure to wildfire-related PM₂.₅, based on a large nationwide dataset in Brazil. The findings were not only representative of the general Brazilian population, but could provide information for assessing the mortality risks and burdens from acute wildfire-related PM₂.₅ exposure in other countries and regions with the similar population demographics, healthcare facilities, and socio-economic status. Consistent evidence suggested associations between wildfire smoke exposure and respiratory diseases; however, evidence on circulatory health is limited. Ours is the first and largest research study to characterize the relationship between exposure to wildfire-related PM₂.₅ and mortality for cardiovascular diseases in Brazil. We also observed a geographic and demographic variations in these associations.

However, this study also has some limitations. Firstly, we only considered wildfire-related PM₂.₅ and its short-term effects on mortality. The potential joint effects of wildfire-related PM₂.₅ and other pollutants, such as ozone and precursor gases, might amplify health effects. Secondly, we could only get access to the exposure estimates for population by using the modeled and spatially refined air pollution data from GEOS-Chem model. Individual exposure representing by region-average tends to be independent of the true exposure level, and to be random[28]. Therefore, our analysis may underestimate the acute impacts of short-term exposure to wildfire-related PM₂.₅. Finally, we could not distinguish between wildfires and deliberately lit fires in this

**Table 2 | Attributable fractions and attributable mortality (with 95% confidence interval [CI]) associated with short-term wildfire-related PM$_{2.5}$ in Brazil during 2000–2016 by cause, sex, age, and region**

| | Attributable fraction (%) | Attributable deaths | Annual attributable rate (per million population) |
|---|---|---|---|
| All-causes | 0.70 (0.41–0.98) | 130,273 (76,534–183,346) | 54.8 (32.2–77.1) |
| Females | 0.88 (0.50–1.24) | 69,803 (40,036–99,068) | 57.2 (32.8–81.1) |
| Males | 0.52 (0.16–0.87) | 55,809 (17,571–93,471) | 48.3 (15.2–80.8) |
| Age 0–59 years | 0.41 (0.00–0.82) | 29,297 (−73–58,170) | 13.7 (0.0–27.3) |
| Age ≥60 years | 0.82 (0.46–1.17) | 93,696 (52,584–134,159) | 383.9 (215.4–549.7) |
| Region | | | |
| North | −0.30 (−1.04–0.42) | −3103 (−10,846–4396) | −47.8 (−167.2–67.8) |
| Northeast | 0.30 (0.11–0.49) | 14,229 (5005–23,371) | 21.6 (7.6–35.5) |
| Southeast | 1.09 (0.88–1.31) | 95,827 (76,919–114,607) | 86.0 (69.0–102.9) |
| South | 0.27 (−0.13–0.67) | 7925 (−3811–19,549) | 21.8 (−10.5–53.8) |
| Central west | 1.31 (0.79–1.82) | 15,395 (9268–21,423) | 87.0 (52.4–121.1) |
| Cardiovascular | 0.63 (0.14–1.10) | 32,961 (7628–57,756) | 13.9 (3.2–24.3) |
| Females | 0.60 (−0.05–1.23) | 15,012 (−1162–30,720) | 12.3 (−1.0–25.2) |
| Males | 0.56 (−0.05–1.15) | 15,383 (−1405–31,729) | 13.3 (−1.2–27.4) |
| Age 0–59 years | 0.46 (−0.48–1.36) | 5351 (−5527–15,784) | 2.5 (−2.6–7.4) |
| Age ≥60 years | 0.64 (0.08–1.19) | 26,277 (3207–48,775) | 107.7 (13.1–199.8) |
| Region | | | |
| North | 0.60 (−1.37–2.40) | 550 (−1263–2210) | −34.7 (−81.7–9.5) |
| Northeast | 0.26 (−0.37–0.88) | 1048 (−1466–3488) | 2.7 (−4.5–9.8) |
| Southeast | 2.29 (1.80–2.78) | 23,127 (18,137–28,041) | 23.6 (16.5–30.7) |
| South | 1.52 (0.51–2.51) | 5084 (1695–8395) | 5.4 (−12.0–22.6) |
| Respiratory | 1.73 (0.98–2.45) | 33,807 (19,225–47,919) | 14.2 (8.1–20.2) |
| Females | 2.11 (1.04–3.13) | 19,324 (9490–28,659) | 15.8 (7.8–23.5) |
| Males | 1.24 (0.18–2.25) | 12,864 (1840–23,398) | 11.1 (1.6–20.2) |
| Age 0–59 years | 0.67 (−1.41–2.57) | 2682 (−5642–10,283) | 1.3 (−2.6–4.8) |
| Age ≥60 years | 1.85 (1.06–2.62) | 28,662 (16,407–40,466) | 117.4 (67.2–165.8) |
| Region | | | |
| North | −1.01 (−2.37–0.28) | −2253 (−5298–619) | 8.5 (−19.5–34.1) |
| Northeast | 0.14 (−0.23–0.51) | 1808 (−2931–6464) | 1.6 (−2.2–5.3) |
| Southeast | 1.02 (0.71–1.33) | 26,345 (18,373–34,241) | 20.8 (16.3–25.2) |
| South | 0.23 (−0.50–0.94) | 1973 (−4375–8213) | 14.0 (4.7–23.1) |
| Central west | 1.55 (0.57–2.51) | 5088 (1859–8219) | 22.6 (12.0–32.7) |

study. We considered wildfires as all fires burning in forest, grassland, bushland, or cropland with the same health impacts from combustion of biomass in wildlands.

Our findings suggested that short-term exposure to wildfire-related PM$_{2.5}$ is associate with increase in mortality risks of all-cause, cardiovascular, and respiratory, even for populations living further away from the burnt areas. We also observed stronger associations among females and older adults aged ≥60 years. These findings have important implications for adaptation strategies and emergency planning to better mitigate wildfire-related health risks under conditions of increasing wildfire risks in Brazil[29]. For example, public health professionals should educate residents raising awareness of wildfire smoke crisis and also guide prompt public responses and take actions to reduce exposure, especially for sensitive populations.

## Methods

### Date sources

We collected death records from 2000 to 2016 from the Brazil Mortality Information System (Sistema de Informação sobre Mortalidade, SIM) that covers the Brazilian population[30]. Each death record included information on individual's municipality, age, sex, death date and primary cause of death coded according to the International Statistical Classification of Diseases and Related Health Problems, 10th Revision (ICD-10). Cardiovascular or respiratory deaths were defined as deaths with primary cause of death coded as I00–I99 or J00–J99, respectively. Daily counts of all-cause, cardiovascular, and respiratory deaths were grouped into 510 Brazilian immediate regions. Daily death counts of each sex (male, female) and age groups (0–59 years, ≥60 years) were also grouped in a same manner. According to the Brazilian Institute of Geography and Statistics (IBGE), immediate regions represents regional geographic divisions linked to social, political and economic processes observed in the national territory, with each immediate region containing several municipalities[31]. Population sizes at municipality-level were downloaded from the website of the Brazilian Institute of Geography and Statistics (BIGS, https://www.ibge.gov.br/pt/inicio.html) and were then grouped into immediate region-levels.

To allow adjustment for the meteorological variables, we collected hourly surface temperature and ambient dew point temperature from the European Centre for Medium-Range Weather Forecasts Reanalysis, v5 (ERA5) at a 0.25° × 0.25° spatial resolution[32]. This dataset has global coverage and is comparable to weather station observations in evaluating temperature-mortality associations[33]. Hourly data were averaged into daily values. We then calculated daily mean relative humidity from the ERA5 daily mean temperature and daily mean dew point temperature, using the algorithm provided by the "humidity" R package[34]. The municipality-level temperature and relative humidity were represented by the value of the grid at the geographical centre of each city[35]. The immediate region-average values of meteorological

factors were then derived from population-weighted averages of all municipality-level values within the region.

## Estimation of wildfire-related PM$_{2.5}$

As detailed in our previous work[10,19,36], daily wildfire-related PM$_{2.5}$ during the study period were estimated as the differences between simulations with or without fire emissions using GEOS-Chem at a spatial resolution of 2.0° latitude × 2.5° longitude. Fire-induced disturbance in PM$_{2.5}$ then was downscaled to generate the ratio of wildfire-related PM$_{2.5}$ to all-source PM$_{2.5}$ at a spatial resolution of 0.25° × 0.25° using inverse distance weighted spatial interpolation. As monitored wildfire-related PM$_{2.5}$ was not available, all-source PM$_{2.5}$ from GEOS-Chem was validated and calibrated against global wide ground-level monitoring PM$_{2.5}$ based on a random forest model with GEOS-Chem generated all-source PM$_{2.5}$ and meteorological variables as predictors. The calibrated all-source PM$_{2.5}$ reached high accuracy compared with ground-based observations (10-fold cross-validation: $R^2 = 86.5\%$, root mean square error = 15.1 μg/m$^3$). Then, the final calibrated wildfire-related PM$_{2.5}$ was derived by multiplying calibrated all-source PM$_{2.5}$ with the downscaled ratio of wildfire-related to all-source PM$_{2.5}$. Details for data estimation, validation and adjustment could be found in our previous work[10].

## Statistical analysis

The PM$_{2.5}$-mortality associations were analyzed using a two-stage time-series approach[37]. In the first stage, we applied a standard quasi-Poisson generalized linear model to examine the immediate region-specific association between daily concentration of wildfire-related PM$_{2.5}$ and cause-specific death counts. A cross-basis function was defined using a linear function for the space of PM$_{2.5}$, and a natural cubic spline for the space of 14 lag days with 4 degrees of freedom (df). We initially compared the lagged patterns of wildfire-related PM$_{2.5}$ on mortality during lags with the maximum of 7, 10, and 14 lag days (Supplementary Fig. 4). We found that a maximum of 14 lag days could fully capture the lagged effects. The model adjusted for the 21-day moving averages of daily mean temperature with 3 df natural cubic spline and 7-day moving average of daily mean relative humidity with 3 df natural cubic spline. Day of week and public holidays presented as categorical variables were also controlled in the models. Seasonality and long-term trends were controlled using a natural cubic spline of time with 7 degrees of freedom per year, a common choice for time-series studies[38]. In the second stage, we pooled the immediate region-specific estimates at the national level by meta-analysis. The lagged effects and cumulative effects of wildfire-related PM$_{2.5}$ on mortality were described as the relative risk (RR) of mortality and corresponding 95% confidence intervals (CIs) per 10 μg/m$^3$ increase in PM$_{2.5}$ concentrations. See Supplementary Fig. 5 for the framework of statistical analysis.

$$Y_{ti} \sim poisson(\mu_{ti})$$

$$Log(\mu_{ti}) = \alpha + cb(PM_{2.5t}, df = 4) + ns(Temp_t, df = 3) + ns(RH_t, df = 3) \\ + ns(Time_t, df = 7 \times n) + DOW_t + Holidays_t \quad (1)$$

Based on the effect estimates, we calculated the burden of mortality attributable to wildfire-related PM$_{2.5}$ as the attributable number of cause-specific deaths (AD) for each immediate region i using previously published methods[39],

$$AD_{ti} = D_{ti} \times (RR_{ti} - 1)/RR_{ti} \quad (2)$$

$$RR_{ti} = exp(\beta_i \times \triangle x_{ti}) \quad (3)$$

where D$_{ti}$ is the immediate region-specific average daily death counts from day t to day t+14 in immediate region t; RR$_{ti}$ is the

cumulative relative risk associated with increase in concentration of wildfire-related PM$_{2.5}$in immediate region i on day t from the above analyses (β$_i$); $\triangle x_{ti}$ is the wildfire-related PM$_{2.5}$ concentration in immediate region i on day t. The 95% CI of AD was calculated replacing RR with its 95% CI bounds. The corresponding attributable fractions (AFs) of mortality and their 95% CIs were calculated by dividing the total AD by total mortality counts, and the corresponding rate of attributable cases per million population (attributed mortality rates, AMRs) and their 95% CIs by dividing the total AD by total population. These analyses were performed separately for all causes, cardiovascular, and respiratory mortalities, and for age and sex subgroups.

To further examine the robustness of results, sensitivity analyses were conducted for the immediate region-specific models. We varied the maximum lag time from 14 to 13, 15, or 16 days for wildfire-related PM$_{2.5}$ or the degrees of freedom of lag days (from 4 to 3 or 5) in the cross-basis function. We also changed the df of meteorological variables (from 3 to 4 or 5) and the moving average days of temperature (from 21-day to 14-day).

All data analyses were performed using R software (version 3.6.1). The dlnm package was used to fit a distributed lag linear model, and the "mvmeta" package to fit meta-analysis and meta- regression[40]. $p$ values < 0.05 (two-sided) were considered as statistically significant

## Reporting summary

Further information on research design is available in the Nature Portfolio Reporting Summary linked to this article.

## Data availability

Publicly available data is found here: population data from the Brazilian Institute of Geography and Statistics (BIGS, https://www.ibge.gov.br/pt/inicio.html); surface temperature and ambient dew point temperature from the European Centre for Medium-Range Weather Forecasts Reanalysis, v5 (ERA5, https://www.ecmwf.int/en/forecasts/datasets/reanalysis-datasets/era5); the base map of Fig. 1A–D from the Brazilian of Geography and Statistics (https://www.ibge.gov.br/en/geosciences/territorial-organization/territorial-meshes/18890-municipal-mesh.html?=&t=o-que-e). Modern-Era Retrospective analysis for Research and Applications version 2 (MERRA-2) data, biomass burning emissions inventory of Global Fire Emissions Database version 4.1 (GFED V4.1) data, and anthropogenic emissions inventory of EDGAR version 4.2 data that support the GEOS-Chem model development and wildfire-related PM$_{2.5}$ simulation in this study are available from https://gmao.gsfc.nasa.gov/reanalysis/MERRA-2/, https://daac.ornl.gov/VEGETATION/guides/fire_emissions_v4_R1.html, and http://edgar.jrc. ec.europa.eu/, respectively. Source data for the figures are provided under this following link: https://github.com/pipty/2022_Brazil_firePM2.5_mortality. For health outcome data from the Brazil Mortality Information System, the authors are not permitted to share the raw data. Researchers who are interested should contact the data provider via https://ghdx.healthdata.org/series/brazil-mortality-information-system-sim. Secondary data (daily death count data) used in the analyses could be shared by contacting the corresponding authors (Yuming Guo: yuming.guo@monash.edu; Shanshan Li: shanshan.li@monash.edu).

## Code availability

The R codes for the epidemiological analyses and R codes for figures are provided under this following link: https://github.com/pipty/2022_Brazil_firePM2.5_mortality.

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

## Acknowledgements

This study was supported by the Australian Research Council (DP210102076) and the Australian National Health and Medical Research Council (GNT2000581). T.Y., R.X., and P.Y. were supported by the China Scholarship Council (grant number 201906320051 for T.Y., 201806010405 for R.X., and 201906210065 for P.Y.); S.L. and Y.G. by Fellowships of the Australian National Health and Medical Research Council (grant number GNT2009866 for S.L., GNT1163693 and GNT2008813 for Y.G.); G.C. by the National Natural Science Foundation of China (grant number 81903279); X.Y. by Jiangsu Science Fund for Distinguished Young Scholars (grant number BK20200040); and MSZSC and PHNS by the São Paulo Research Foundation.

## Author contributions

Y.G. and S.L. conceived, designed, and supervised the study. T.Y. analyzed the data and prepared the first draft. X.Y. performed the GEOS-Chem simulations. G.C., R.X., and T.Y. prepared exposure data. R.X., P.Y., P.H.N.S., and M.S.Z.S.C. prepared the health database and did the quality assurance. Y.G., S.L., and T.Y. led the drafting of the manuscript, interpretation of the results, and verification of the underlying data. All authors reviewed and edited the paper.

## Competing interests

M.J.A. holds investigator initiated grants for unrelated research from Pfizer, Boehringer-Ingelheim, Sanofi and GSK. He has undertaken an unrelated consultancy for and received assistance with conference attendance from Sanofi. He has also received a speaker's fee from GSK. The other authors declare no conflicts of interest.
