## [Peer Review File · Nature Communications]

Reviewers' Comments:

Reviewer #1:

Remarks to the Author:

Review of manuscript submitted for publication in Nature Comm

Short-term exposure to wildfire fine particulate matter and mortality risks and burdens in Brazil: A nationwide time-series study between 2000 and 2016

This manuscript deals with a relevant issue, with a very large dataset of mortality in Brazil (18,681,906 records). But I have some difficulties with the methodology and the lack of really new findings in terms of scientific issues. I think the main positive issue with this manuscript is tentative to get the association between wildfire-specific PM_{2.5} and the risk of deaths from specific causes (e.g., respiratory, cardiovascular causes). They also tried to separate total PM_{2.5} from wildfire PM_{2.5}. The methodology uses ECMWF reanalysis, and they claim that the dataset has global coverage and is comparable to weather station observations in evaluating temperature-mortality associations. BUT this could be true for temperature and is certainly not valid for PM_{2.5}.

The authors claim that "daily wildfire-related PM_{2.5} during the study period was estimated as the differences between simulations with or without fire emissions using GEOSChem at a spatial resolution of 2.0° latitude × 2.5° longitude. Fire-induced disturbance in PM_{2.5} then was downscaled to generate the ratio of wildfire-related PM_{2.5} to all-source PM_{2.5} at a spatial resolution of 0.25° × 0.25° using interpolation". The spatial resolution of 2.0° latitude × 2.5° longitude is far too coarse to be used for PM_{2.5} estimation for wildfires. If you do downscale through interpolation, I think it is not an accepted procedure, because the original data is far too coarse for many municipalities. Additionally, the GEOSChem simulation was not validated for Amazonian conditions. Removal rates due to precipitation or transport not accurately predicted by the model could affect significantly the analysis. I think the authors need to provide stronger evidence that the methodology works in their situation and analysis. Why not use remote sensing of PM_{2.5}? Copernicus and NASA provide PM_{2.5} data with a spatial resolution of 10 or 50 Km. Or at least validate the GEOS-Chem with satellite OPM_{2.5} measurements.

The methodology to get wildfire emissions with two runs, with and without fires, is questionable. Especially because the analysis was done for the whole of Brazil, including areas with no wildfires. There is a strong seasonality of wildfires. How was this treated in the study? Also, the authors do not show the geographical distribution of wildfire mortality. It must be much higher in Amazonia, and much lower in the Rio Grande do Sul, for instance. The authors fail in this important discussion of spatial and temporal variability in their findings.

I think the manuscript requires really major revisions. It certainly needs to be much improved to be accepted for publications in Nature Comm.

Specific issues:

In lines 58-59, the authors conclude that "Actions to save energy and cut carbon emissions would limit global temperature increase and mitigate wildfire risks". The link between "save energy" is not linked with wildfire risks or occurrence.

Line 62: Manuscript says that "Wildfires have become an increasingly visible and potent threat in Brazil, modulated by both climate and human-driven land-use changes". Wildfires occur not only in Brazil, and wildfires are all human-driven in Brazil. Climate or lightning does not play a role in wildfires in Amazonia.

Figure 1 is strange: Why only 4 classes? Why all above 3.51 is on the same range? This does not make sense for the type of analysis being done in this manuscript.

Lines 295 to 304. This whole paragraph is dedicated to showing that the results of this study are in the range of other 6 different studies. The issue is: What exactly this study advances the sciences in this area? The two last paragraphs are dedicated to answering this question, but they do not really address this critical issue.

Reviewer #2:

Remarks to the Author:

Dear Authors and Editors

Thank you for the opportunity to review this work. Your hypothesis is important and your work provides noteworthy results on the risk of mortality in the country very much vulnerable to smoke exposure. I think the manuscript was well written and clear and I only have two comments I would like to note before proceeding to comment and provide full review. On line 165 you say that the effects were described as the relative rr. Can you clarify for me how that was done. The second comment is to clarify the equation on line 171. What does EE stand for?

I really appreciate your responsiveness to this request because it has an impact of how well I can evaluate whether the methodology was sound and interpret results.

Reviewer #3:

Remarks to the Author:

The manuscript 'Short-term exposure to wildfire 1 fine particulate matter and mortality risks 2 and burdens in Brazil: A nationwide time-series study between 2000 and 2016' addresses a critical and timely environmental pressure. In particular in the light of the recently published UNEP report "Spreading like Wildfire: The Rising Threat of Extraordinary Landscape Fires" <https://www.unep.org/resources/report/spreading-wildfire-rising-threat-extraordinary-landscape-fires>, health impacts from wildfire smoke in regions where this topic has so far not received sufficient attention need to be assessed and this study makes a vital contribution to the scientific discourse on this matter.

I have very few comments, which are primarily suggestions on small improvements, and have not identified any substantive flaws or issues throughout the well-written and clearly presented manuscript.

Introduction

L68ff: I interpret the concentrations referred to in this section as total PM_{2.5} of 33 and up to 100 µg_m⁻³ respectively, and not the contribution from fire emissions, however, this could potentially be made more clear. Equally in this section, if (!) there is any information about the composition of PM_{2.5} available in any of the regions investigated, giving a relative contribution of fire vs. non-fire emissions would be very informative, however, this data may not be available.

Methods

L124ff: More a question out of curiosity, is there a potential effect in the temporal correlation with high temperature and increased fire activity, as well as high temperature and mortality risk, which may amplify mortality results?

L136ff: Related to the comment above on speciated observations, I wonder if a short paragraph evaluating the relative agreement between GEOS-Chem modelled PM_{2.5} composition and some spot measurements (e.g. from studies such as <https://www.ncbi.nlm.nih.gov/pmc/articles/PMC4448729/> or <https://acp.copernicus.org/articles/17/11943/2017/acp-17-11943-2017.pdf>) could add further confidence in the robustness of the results.

This ties in as well with the notable spatial variability of concentrations vs. mortality rates illustrated in Figure 1: Ambient concentrations from wildfire related PM_{2.5} in panel are clearly most pronounced in the central-Western region, whereas hotspot of mortality in general are focused on the Northeast and South/Southeast regions. There are various options for explaining this, one could be the relative contribution of wildfire related PM_{2.5} to already high levels of particulate matter pollution in densely populated urban centres vs. less dense populations and hence reduced exposure in comparatively 'cleaner' rural environments. Potentially adding a third panel with the population distribution could add further clarity to Figure 1?

A small editorial comment on Figure 1, the caption is not entirely clear, as panel A appears to show ambient average concentrations of PM_{2.5} in $\mu\text{g m}^{-3}$, but the caption is referring to mortality rates/100,000 population? Please check and clarify?

Fig 3. I may have missed it, but the significance of the colored dots in the figure are not explained in the caption or elsewhere? If they are simply related to grouping e.g. m/f, age groups, regions, the colors may not be needed?

L296ff: the difference between exposure to wildfire-related PM_{2.5} vs a scenario where no wildfire-related PM_{2.5} exposure would occur could be simulated with GEOS-Chem - if such simulations have indeed been undertaken, presenting difference maps between with and without fire emissions for the period from 2000 - 2016 would be a very valuable addition to the paper, potentially as suppl. mat.? This could illustrate as well without much additional interpretation needed the interannual variability, as well as a potential trend between PM_{2.5} generated by wildfire activity vs other sources (see next comment).

L323ff: With the long timescale of data assessed, have any confounding factors been considered e.g. in the case that wildfire PM_{2.5} may have had an increasing trend, while the contribution of other (anthropogenic) sources from transport or stationary combustion may have slightly decreased over time? Such underlying trends could be identified by the comparative model assessment suggested above, but may be beyond the scope of this paper.

Conclusions

The UNEP Report referenced in my introductory comments was released after the manuscript was submitted, but a short additional reference and a couple of sentences reflecting on what the findings of this manuscript would mean under conditions of increasing wildfire risks due to climate change in the region may be worth adding, to further increase the potential impact of the paper.

All of the comments above are primarily suggestions, not identifying substantial flaws or shortcomings of the manuscript as is!

REVIEWER COMMENTS

Reviewer #1

This manuscript deals with a relevant issue, with a very large dataset of mortality in Brazil (18,681,906 records). But I have some difficulties with the methodology and the lack of really new findings in terms of scientific issues. I think the main positive issue with this manuscript is tentative to get the association between wildfire-specific PM_{2.5} and the risk of deaths from specific causes (e.g., respiratory, cardiovascular causes). They also tried to separate total PM_{2.5} from wildfire PM_{2.5}. The methodology uses ECMWF reanalysis, and they claim that the dataset has global coverage and is comparable to weather station observations in evaluating temperature-mortality associations. BUT this could be true for temperature and is certainly not valid for PM_{2.5}.

The authors claim that “daily wildfire-related PM_{2.5} during the study period was estimated as the differences between simulations with or without fire emissions using GEOSChem at a spatial resolution of 2.0° latitude × 2.5° longitude. Fire-induced disturbance in PM_{2.5} then was downscaled to generate the ratio of wildfire-related PM_{2.5} to all-source PM_{2.5} at a spatial resolution of 0.25° × 0.25° using interpolation”. The spatial resolution of 2.0° latitude × 2.5° longitude is far too coarse to be used for PM_{2.5} estimation for wildfires. If you do downscale through interpolation, I think it is not an accepted procedure, because the original data is far too coarse for many municipalities. Additionally, the GEOSChem simulation was not validated for Amazonian conditions. Removal rates due to precipitation or transport not accurately predicted by the model could affect significantly the analysis. I think the authors need to provide stronger evidence that the methodology works in their situation and analysis. Why not use remote sensing of PM_{2.5}? Copernicus and NASA provide PM_{2.5} data with a spatial resolution of 10 or 50 Km. Or at least validate the GEOS-Chem with satellite OPM_{2.5} measurements.

The methodology to get wildfire emissions with two runs, with and without fires, is questionable. Especially because the analysis was done for the whole of Brazil, including areas with no wildfires. There is a strong seasonality of wildfires. How was this treated in the study? Also, the authors do not show the geographical distribution of wildfire mortality. It must be much higher in Amazonia, and much lower in the Rio Grande do Sul, for instance. The authors fail in this important discussion of spatial and temporal variability in their findings.

I think the manuscript requires really major revisions. It certainly needs to be much improved to be accepted for publications in Nature Comm.

[Response] We address your concerns point-by-point.

(1) Estimation of wildfire-related PM_{2.5} using GEOS-Chem model is a proven method.

GEOS-Chem is a widely used chemical transport model that has been applied to estimate the contribution to ambient air pollutant concentrations from specific emission sectors^[1]. McDuffie et al. used this method to quantify the relative contributions from 24 individual emission sectors globally^[2].

(2) Epidemiological studies have used a similar approach to assess fire PM_{2.5} and its health impacts^[3-5]. Further details of the method could be found in our previous work^[1] and the supplemental material of our latest publication^[6].

(3) Validation for Amazonian conditions is unavailable.

We acknowledge the uncertainties in exposure assessment due to PM_{2.5} simulations by GEOS-Chem model, because regional emission factors were not included. A previous study from the South American Biomass Burning Analysis (SAMBBA) experiment in the Amazon showed good agreement between observed and modelled aerosol optical depth (AOD) by GFED3 emissions was gained after multiplying by a factor of 1.6 to 2 to arrive at real emissions^[7], ie. the model underestimated emissions. There are important uncertainties in fire activities and emission simulation, regarding the emission factors of fires and models. GFED4.1 now included in GEOS-Chem is the most up-to-date available emissions system (GFED4). Although underestimation of real concentrations may be possible into some extent, it would occur for periods with or without fires, thus having less influence on the observed differences. In addition, six different types of biomes are defined in Brazil: Amazon, Atlantic Forest, Cerrado, Caatinga, Pampa and Pantanal (<https://organicbrasil.org/brasil-biomes/>). All biomes are diverse in climate, culture, agriculture and soil management and seasonal wildfires. In other words, Brazil should not be generalized from the Amazonian information and database.

(4) Simulation of wildfire-related PM_{2.5} with coarse and fine resolutions have very similar spatial patterns and would not affect the main conclusions.

In the GEOS-Chem model, PM_{2.5} concentrations are computed at 35% relative humidity (RH), which accounts for water uptake consistent with PM_{2.5} measurements (http://wiki.seas.harvard.edu/geos-chem/index.php/Particulate_matter_in_GEOS-Chem). The mass of PM_{2.5} is calculated based on ammonium aerosol (NH₄), inorganic nitrate aerosol (NIT), sulfate aerosol (SO₄), hydrophilic black carbon aerosol (BCPI), hydrophobic black carbon aerosol (BCPO), hydrophobic organic carbon aerosol (OCPO), hydrophilic organic carbon aerosol (OCPI), secondary organic aerosol (SOA), DUST1, DUST2, accumulation mode of sea salt aerosol (SALA), aerosol-phase glyoxal (SOAGX), aerosol-phase methylglyoxal (SOAMG) following the Equation:

$$\text{PM}_{2.5} = 1.33 (\text{NH}_4 + \text{NIT} + \text{SO}_4) + \text{BCPI} + \text{BCPO} + 2.10 (\text{OCPO} + 1.16 \text{OCPI}) + 1.16 \text{SOA} + \text{DUST1} + 0.38 \text{DUST2} + 1.86 \text{SALA} + \text{SOAGX} + \text{SOAMG}$$

We performed additional sensitivity tests to explore the impacts of spatial resolution on simulated fire PM_{2.5} concentrations. The finest resolution for the global simulation in GEOS-Chem is 2°×2.5°, which was applied in this study. Regionally, the GEOS-Chem model provides nested grids of 0.5°×0.625° over Asia and North America. Because of the high computational costs, we conducted four short-term tests for a randomly selected period of April-August 2015. Two runs were performed over Asia with or without fire emissions. The other two were performed over North America with or

without fire emissions. Simulated $PM_{2.5}$ concentrations, no matter at low (original runs) or high (new runs) resolutions, were interpolated to the locations of cities where the observed $PM_{2.5}$ and health data have been used. In total, there were 170 cities in Asia and 247 in North America.

We compared the fire-induced monthly $PM_{2.5}$ concentrations from low and high resolutions during June-August 2015 (Figure R1). High correlation coefficients of 0.92-0.94 were derived between the simulations with low ($2^\circ \times 2.5^\circ$) and high ($0.5^\circ \times 0.625^\circ$) resolutions over two nested regions, suggesting that simulations with two different resolutions retained similar spatial characteristics. The high-resolution runs showed lower concentrations than the low-resolution runs, but the differences were within 15%. Regional sensitivity tests with a higher resolution of $0.5^\circ \times 0.625^\circ$ show very similar spatial distribution of fire $PM_{2.5}$ with a mean difference within 15%, suggesting that the coarse model resolution would not affect the main conclusions of this study.

Figure R1. Comparisons of monthly fire $PM_{2.5}$ between nested ($0.5^\circ \times 0.625^\circ$) and global simulations ($2^\circ \times 2.5^\circ$) over Asia (a) and North America (b) during 2015 summer. Each point represents monthly fire $PM_{2.5}$ at one site. The fire $PM_{2.5}$ is derived as the differences of $PM_{2.5}$ between simulations with and without fire emissions. The red lines indicate linear regressions between predictions from nested and global simulations. The correlation coefficient (R), normalized mean biases (NMBs), and sample number (N) are shown in each panel.

(5) Downscaled total $PM_{2.5}$ from our simulation showed a high cross validation R square of 86.5% with the ground measured $PM_{2.5}$.

As wildfire-related $PM_{2.5}$ was not routinely monitored and their measurements were unavailable in Brazil, we compared GEOS-Chem-derived estimates of all-source $PM_{2.5}$ with a global dataset of ground-level measurements from 6,882 monitoring sites using a random forest model. Details on the random forest model and cross-validation could be found from the supplemental material in our latest publication^[6]. The results showed that “The calibrated all-source $PM_{2.5}$ reached high accuracy compared with ground-based observations (10-fold cross-validation: R square = 86.5%, root mean square error = $15.1 \mu\text{g}/\text{m}^3$) (Line 143-146)”

(6) All months experienced wildfire smoke in Brazil

This study assessed the mortality risks associated with $PM_{2.5}$ from wildfire smoke, which were beyond the direct risks from exposure to fires or involvement in wildfire events. $PM_{2.5}$ is the major pollutant of public health concern during wildfire events, which could spread as far as 1000 km away^[8] and increase risks of illness and death far away from the burning sites.

Figure R2 shows that the mean daily wildfire-related $PM_{2.5}$ varied across months and years. Actually, all months experienced wildfire smoke. In addition, fires in the Amazon shift with season and regions year-to-year, for example, fires occurred early in the year in the northern Amazon, while fires in the southern Amazon began in July and peaked in August and September (<https://rainforestfoundation.org/fires-in-the-amazon-shift-with-seasons-rage-on/>).

Figure R2. Daily wildfire-related $PM_{2.5}$ in different months (A) and years (B).

(7) The spatial distribution of attributable mortality burden

The fraction and rate of all-cause mortality attributable to wildfire-related PM_{2.5} over lag 0-14 days of exposure in 510 Brazilian immediate regions during 2000-2016 has been shown in Figure R3. The estimated attributable mortality was the highest in the central west and followed by the southeast region.

Figure R3. The fraction (A) and rate (B) of all-cause mortality attributable to wildfire-related PM_{2.5} over lag 0-14 days of exposure in 510 Brazilian immediate regions during 2000-2016. The cumulative association was used to calculate the attributable burden.

Specific issues:

In lines 58-59, the authors conclude that “Actions to save energy and cut carbon emissions would limit global temperature increase and mitigate wildfire risks”. The link between “save energy” is not linked with wildfire risks or occurrence.

[Response] We meant to say that carbon emissions could be reduced, by switching to cleaner energy sources. To make it clearer, we modified the sentence as, “Actions to cut carbon emissions would limit global temperature increase and mitigate wildfire risks” (Line 58, clean version).

Line 62: Manuscript says that “Wildfires have become an increasingly visible and potent threat in Brazil, modulated by both climate and human-driven land-use changes”. Wildfires occur not only in Brazil, and wildfires are all human-driven in Brazil. Climate or lightning does not play a role in wildfires in Amazonia.

[Response] Wildfires are definitely a global concern for human health and safety. This manuscript focuses on the risks of wildfire-related air pollution in the context of Brazil. As reviewer stated, the conditions of Amazonia may not lead lightning strikes to “start” a fire. The wildfires in this area are

mostly caused by human activities. However, it should not be concluded that climate does not shape fire regimes. Climate-driven fire risk in the Brazilian Amazon is primarily controlled by two main climate variables: rainfall and temperature^[9]. Increases in temperatures highlight the vulnerability of the Amazon region in a warming climate where increased frequency and intensity of heat waves provide more ignition sources for wildfires. “A number of severe drought episodes have occurred during 2005, 2007 and 2010, and fire emissions increased 1.5 to 2.8 fold compared with non-drought years^[10].” (Line 65, clean version)

Figure 1 is strange: Why only 4 classes? Why all above 3.51 is on the same range? This does not make sense for the type of analysis being done in this manuscript.

[Response] Thanks for this comment.

We used daily data of wildfire-related PM_{2.5} and cause-specific mortality to analyze the associations. It would be less representative to plot the concentration of wildfire-related PM_{2.5} and daily mortality rate on a specific day. Thus, we plotted the annual average values to show the spatial patterns. In Figure 1, we grouped immediate regions by quartile to show the spatial variation of wildfire-related PM_{2.5} (panel A) and all-cause mortality rates (panel B) across Brazil. The cutpoint of 3.51 µg/m³ represents that 25% proportion of the immediate regions had an annual average wildfire-related PM_{2.5} above 3.51.

What's more, classifying the immediate regions into multiple classes as in Figure 1 (revised version) does not alter the spatial patterns.

Figure 1 (revised version). Regional differences in yearly average wildfire-related PM_{2.5} (A), yearly average total PM_{2.5} (B), annual population (C), and mortality rates/100,000 population/year (D) across 510 immediate regions in Brazil during 2000-2016.

Lines 295 to 304. This whole paragraph is dedicated to showing that the results of this study are in the range of other 6 different studies. The issue is: What exactly this study advances the sciences in this area? The two last paragraphs are dedicated to answering this question, but they do not really address this critical issue.

[Response] There are several strengths in this study. First, our estimates are robust due to the use of a large nationwide time-series data in Brazil from 200 to 2016. “Our study linked mortality records covering the entire population in Brazil across 510 immediate regions with estimated daily exposure to wildfire-related PM_{2.5} over 17 years. The findings were representative of the general Brazilian population, and could provide information for assessing the mortality risks and burdens

from acute wildfire-related PM_{2.5} exposure in other countries and regions with the similar population demographics, healthcare facilities, and socioeconomic status.” (Lines 319-321, clean version). Second, ours is the first and largest research study to characterize the relationship between exposure to wildfire-related PM_{2.5} and age- and sex-, cause-specific mortality for respiratory and cardiovascular diseases in Brazil and to examine the geographic and demographic variations in the association. Age- and sex-stratified analyses suggested that “Females and those aged 60 years or older appeared to be more sensitive to wildfire-related PM_{2.5} exposure in Brazil (Line 276, clean version)”. The findings of our study provide an additional perspective when considering public health protection policy. We also rewrote the lines 321-332 (clean version) in the manuscript to highlighted the strengths of this study.

Please also refer to the third paragraph of the Introduction section where we summarized limitations in the previous studies, as “However, they only used total particulate matter concentrations during wildfire events and the risk of death from all-cause or non-accidental deaths. The association between wildfire-specific PM_{2.5} and the risk of deaths from specific causes (e.g., respiratory, cardiovascular causes) remained uncertain. Existing evidence was limited to single-city or single-region near the location of fire sources during burning seasons due to known consequences of biomass burning and emissions. Little was known about the magnitude and scope of its adverse effects on air quality and public health across the nation and whether certain populations were more susceptible (Line 91-99, clean version).”

Reviewer #2

Dear Authors and Editors

Thank you for the opportunity to review this work. Your hypothesis is important and your work provides noteworthy results on the risk of mortality in the country very much vulnerable to smoke exposure. I think the manuscript was well written and clear and I only have two comments I would like to note before proceeding to comment and provide full review. On line 165 you say that the effects were described as the relative rr. Can you clarify for me how that was done. The second comment is to clarify the equation on line 171. What does EE stand for?

I really appreciate your responsiveness to this request because it has an impact of how well I can evaluate whether the methodology was sound and interpret results.

[Response] We are sorry for any unclear statements. We have improved the Methods section in the revision, and also provide a detailed statement in the supplementary material.

As stated in our previous study, the PM_{2.5}-mortality associations were analyzed using a two-stage time-series approach.

In the first stage, we applied a standard quasi-Poisson generalized linear model to examine the immediate region-specific association between daily concentration of wildfire-related PM_{2.5} and cause-specific death counts. A cross-basis function was defined using a linear function for the space of PM_{2.5}, and a natural cubic spline for the space of 14 lag days with 4 degrees of freedom (df). The model adjusted for the 21-day moving averages of daily mean temperature with 3 df natural cubic spline and 7-day moving average of daily mean relative humidity with 3 df natural cubic

spline. Day of week and public holidays presented as categorical variables were also controlled in the models. Seasonality and long-term trends were controlled using a natural cubic spline of time with 7 degrees of freedom per year, a common choice for time-series studies in this field.

$$\log(\mu_t) = cb (PM_{2.5t}, df = 4) + ns (Temp_t, df = 3) + ns(RH_t, df = 3) + ns (Time_t, df = 7 * n) + DOW_t + Holidays_t$$

In the second stage, we pooled the immediate region-specific estimates at the national level by meta-analysis. The lag-response relationship was defined as relative risk (RR) of mortality at the 10 $\mu\text{g}/\text{m}^3$ of wildfire-related $\text{PM}_{2.5}$ concentration comparing to minimum exposure (we assumed that there was no safe threshold level for wildfire-related $\text{PM}_{2.5}$ exposure).

Based on the pooled effect estimates, we calculated the burden of mortality attributable to wildfire-related $\text{PM}_{2.5}$ as the attributable number of cause-specific deaths (AD) using previously published methods,

$$AD_i = D_i \times (RR_i - 1) / RR_i$$

where D_i is the immediate region-specific average daily death counts from day i to day $i+14$; RR_i is the cumulative relative risk associated with increase in concentration of wildfire-related $\text{PM}_{2.5}$ from the above analyses. RR_i is a region-specific effect where the immediate region is located.

Reviewer #3

The manuscript 'Short-term exposure to wildfire fine particulate matter and mortality risks and burdens in Brazil: A nationwide time-series study between 2000 and 2016' addresses a critical and timely environmental pressure. In particular in the light of the recently published UNEP report "Spreading like Wildfire: The Rising Threat of Extraordinary Landscape Fires" <https://www.unep.org/resources/report/spreading-wildfire-rising-threat-extraordinary-landscape-fires>, health impacts from wildfire smoke in regions where this topic has so far not received sufficient attention need to be assessed and this study makes a vital contribution to the scientific discourse on this matter.

I have very few comments, which are primarily suggestions on small improvements, and have not identified any substantive flaws or issues throughout the well-written and clearly presented manuscript.

[Response] We thank the reviewer for the positive feedback on our study.

Introduction

L68ff: I interpret the concentrations referred to in this section as total $\text{PM}_{2.5}$ of 33 and up to 100 $\mu\text{g}/\text{m}^3$ respectively, and not the contribution from fire emissions, however, this could potentially be made more clear. Equally in this section, if (!) there is any information about the composition of $\text{PM}_{2.5}$ available in any of the regions investigated, giving a relative contribution of fire vs. non-fire emissions would be very informative, however, this data may not be available.

[Response] The previous study did not separate fire induced $\text{PM}_{2.5}$ from fire emissions. The numbers of 33 and up to 100 $\mu\text{g}/\text{m}^3$ were referred to concentrations of total $\text{PM}_{2.5}$ during a fire event. To make it clearer, we modified the Line 68 as "regional mean concentrations of total fine

particulate matter with diameters $\leq 2.5\mu\text{m}$ ($\text{PM}_{2.5}$) in heavily impacted sites in south-western Amazon can exceed $33 \mu\text{g}/\text{m}^3$ with daily mean peak concentrations close to $100 \mu\text{g}/\text{m}^3$.”

We were only able to separate wildfire-related $\text{PM}_{2.5}$ from total emissions, while unable to analyse the exact composition of $\text{PM}_{2.5}$ with the current simulation model. We added a map panel in Figure 1 to show the spatial distribution of wildfire-related $\text{PM}_{2.5}$ and total $\text{PM}_{2.5}$ during the study period. In addition, we extracted the gridded fractional source contributing to $\text{PM}_{2.5}$ in Brazil from the 2021 Global Burden of Disease-Major Air Pollution Sources Study-Global (GBD_MAPS)^[2] and added a map panel (Figure R4) to show the annual concentration of source-specific $\text{PM}_{2.5}$. The spatial pattern of simulated wildfire-related $\text{PM}_{2.5}$ by our study (Figure 1, panel A) was similar to the distribution of $\text{PM}_{2.5}$ from open fires estimated by GBD_MAPS with higher concentrations in central west regions. Relatively high concentrations of total $\text{PM}_{2.5}$ in the northern region (Figure 1 revised version, panel B) could be explained by $\text{PM}_{2.5}$ from windblown dust and other remaining sources (“Other” in Figure R4). “Other” sources are primarily from non-combustion or uncategorized combustion sources (agriculture, solvents, biogenic SOA, waste incineration, etc.). We think this may explain the high concentration of total $\text{PM}_{2.5}$ in these regions - as part of the Amazon rainforest- where dead plants and animal matter accumulate continuously.

Figure R4. Annual concentrations of source-specific PM_{2.5} in Brazil for the year 2017. (Data source: 2021 Global Burden of Disease-Major Air Pollution Sources Study-Global^[2])

Methods

L124ff: More a question out of curiosity, is there a potential effect in the temporal correlation with high temperature and increased fire activity, as well as high temperature and mortality risk, which may amplify mortality results?

[Response] Climate-driven fire risk in the Brazilian Amazon is primarily controlled by two main climate variables: rainfall and temperature^[9]. Increase in temperatures highlights the vulnerability of

the Amazon region during dry periods where increased frequency and intensity of heat waves provide flammable vegetation that can fuel wildfires. Higher temperatures increased dry-season length^[11].

The associations between high temperature and mortality/morbidity has been well documented^[12], and heat waves potentially amplify effects of wildfire-related PM_{2.5}. Thus, the confounding effects of mean temperature and relative humidity were controlled, as stated in the manuscript, “The model adjusted for the 21-day moving averages of daily mean temperature with 3 df natural cubic spline and 7-day moving average of daily mean relative humidity with 3 df natural cubic spline (Lines 156-158, clean version).”

L136ff: Related to the comment above on speciated observations, I wonder if a short paragraph evaluating the relative agreement between GEOS-Chem modelled PM_{2.5} composition and some spot measurements (e.g. from studies such as <https://www.ncbi.nlm.nih.gov/pmc/articles/PMC4448729/> or <https://acp.copernicus.org/articles/17/11943/2017/acp-17-11943-2017.pdf>) could add further confidence in the robustness of the results.

[Response] We appreciate the reviewer’s suggestion very much. We were unable to simulate the exact chemical constituents of PM_{2.5} by GEOS-Chem model at present. As the spot measurement data you referred to in these two studies were not open-source, we have contacted the authors for a data request. To be specific, data generated by Rodriguez et al^[13] were total PM_{2.5} from all sources collected in urban, industrial and rural sites in Rio de Janeiro. These could possibly be compared with total simulated PM_{2.5}. And the study conducted by Pereira estimated the chemical composition and source of particulate pollutants in São Paulo, including biomass burning. It could be possible to evaluate the agreement between simulated wildfire-related PM_{2.5} and spot measurements.

As stated, the simulated total PM_{2.5} and wildfire-related PM_{2.5} by GEOS-Chem model could have a high spatial agreement with the source-specific PM_{2.5} from the 2021 Global Burden of Disease-Major Air Pollution Sources Study-Global (GBD_MAPS)^[2] (Figure R4). Due to the scarcity of ground level monitoring stations in Brazil, we included a global dataset with daily monitored PM_{2.5} for cross validation and their differences were further used to adjust the GEOS-Chem-derived wildfire-related PM_{2.5}. The results showed that “The calibrated all-source PM_{2.5} reached high accuracy compared with ground-based observations (10-fold cross-validation: R square = 86.5%, root mean square error = 15.1 µg/m³) (Lines 143-146, clean version)”

This ties in as well with the notable spatial variability of concentrations vs. mortality rates illustrated in Figure 1: Ambient concentrations from wildfire related PM_{2.5} in panel are clearly most pronounced in the central-Western region, whereas hotspot of mortality in general are focused on the Northeast and South/Southeast regions. There are various options for explaining this, one could be the relative contribution of wildfire related PM_{2.5} to already high levels of particulate matter pollution in densely populated urban centres vs. less dense populations and hence reduced exposure in comparatively ‘cleaner’ rural environments. Potentially adding a third panel with the population distribution could add further clarity to Figure 1?

[Response] We have added a map panel (Figure 1 revised version, panel C) with the annual population distribution. There are densely populated areas along the coastal line and hotspots in other regions.

A small editorial comment on Figure 1, the caption is not entirely clear, as panel A appears to show ambient average concentrations of PM_{2.5} in $\mu\text{g m}^{-3}$, but the caption is referring to mortality rates/100,000 population? Please check and clarify?

[Response] Sorry for this typo. We have corrected Figure 1 as above.

Fig 3. I may have missed it, but the significance of the colored dots in the figure are not explained in the caption or elsewhere? If they are simply related to grouping e.g. m/f, age groups, regions, the colors may not be needed?

[Response] Thanks for your suggestion. We have removed the colors.

Figure 3 (revised version). Pooled relative risks of all-cause, cardiovascular, and respiratory mortality (stratified by sex and age groups) associated with a 10 $\mu\text{g}/\text{m}^3$ increase in wildfire-related PM2.5 over lag 0–14 days. (Note: * p value for the differences in cumulative relative risks (with 95% CI) across population subgroups were estimated by fixed effect meta-regression.)

L296ff: the difference between exposure to wildfire-related PM2.5 vs a scenario where no wildfire-related PM2.5 exposure would occur could be simulated with GEOS-Chem - if such simulations have indeed been undertaken, presenting difference maps between with and without fire emissions for the period from 2000 - 2016 would be a very valuable addition to the paper, potentially as suppl. mat.? This could illustrate as well

without much additional interpretation needed the interannual variability, as well as a potential trend between PM_{2.5} generated by wildfire activity vs other sources (see next comment).

[Response] We added a map panel in Figure 1 to present the annual concentration of fire-related PM_{2.5} and total PM_{2.5} for the study period from 2000-2016.

Please see Figure R4 presenting the temporal variability of wildfire-related PM_{2.5} concentrations. There were significant seasonal trends of daily concentrations (please see Figure R2, panel A), while no yearly trend was observed over this period (please see Figure R2, panel B). In the analytical model assessing PM_{2.5}-mortality associations, “Seasonality and long-term trends were controlled using a natural cubic spline of time with 7 degrees of freedom per year, a common choice for time-series studies (Lines 159-161, clean version).”

L323ff: With the long timescale of data assessed, have any confounding factors been considered e.g. in the case that wildfire PM_{2.5} may have had an increasing trend, while the contribution of other (anthropogenic) sources from transport or stationary combustion may have slightly decreased over time? Such underlying trends could be identified by the comparative model assessment suggested above, but may be beyond the scope of this paper.

[Response] There were changes of annual average PM_{2.5} concentration year-to-year, however no increasing/decreasing trends were observed (Figure R5). As we stated in the manuscript, we applied a standard quasi-Poisson generalized linear model to examine the immediate region-specific association between daily concentration of wildfire-related PM_{2.5} and cause-specific death counts. Seasonality and long-term trends were controlled using a natural cubic spline of time with 7 degrees of freedom per year, a common choice for time-series studies in this field (Line 156-161, clean version).

$\log(\mu_t) = \text{cb}(\text{PM}_{2.5t}, \text{df} = 4) + \text{ns}(\text{Temp}_t, \text{df} = 3) + \text{ns}(\text{RH}_t, \text{df} = 3) + \text{ns}(\text{Time}_t, \text{df} = 7 * n) + \text{DOW}_t + \text{Holidays}_t$

Figure R5. Trend of annual average concentrations of total and wildfire-related PM_{2.5} from 2000 to 2016.

Conclusions

The UNEP Report referenced in my introductory comments was released after the manuscript was submitted, but a short additional reference and a couple of sentences reflecting on what the findings of this manuscript would mean under conditions of increasing wildfire risks due to climate change in the region may be worth adding, to further increase the potential impact of the paper.

[Response] We thank the reviewer for this suggestion. We inserted a reference to the UNEP report and rewrote the Conclusion section (Lines 344, clean version), as: “A new report by UNEP finds that climate change and land-use change are making wildfires worse even in areas previously unaffected^[14]. This study confirmed that populations living further away from the burnt areas could be influenced by wildfire-related PM_{2.5} with increased attributable risks and mortality burdens from all-causes, cardiovascular and respiratory in Brazil. In addition, females and those aged 60 years or older were more sensitive to wildfire-related PM_{2.5} than males and the young, respectively. These findings have important implications for developing area-, cause-, and group-specific adaptation strategies and emergency planning to better mitigate wildfire-related health risks.”

References:

1. Yue, X. and N. Unger, *Fire air pollution reduces global terrestrial productivity*. Nat Commun, 2018. **9**(1): p. 5413.
2. McDuffie, E.E., et al., *Source sector and fuel contributions to ambient PM_{2.5} and attributable mortality across multiple spatial scales*. Nat Commun, 2021. **12**(1): p. 3594.
3. Stowell, J.D., et al., *Associations of wildfire smoke PM_{2.5} exposure with cardiorespiratory events in Colorado 2011-2014*. Environ Int, 2019. **133**(Pt A): p. 105151.
4. Xue, T., et al., *Open fire exposure increases the risk of pregnancy loss in South Asia*. Nat Commun, 2021. **12**(1): p. 3205.
5. Liu, J.C., et al., *Wildfire-specific Fine Particulate Matter and Risk of Hospital Admissions in Urban and Rural Counties*. Epidemiology, 2017. **28**(1): p. 77-85.
6. Chen, G., et al., *Mortality risk attributable to wildfire-related PM_{2.5} pollution: a global time series study in 749 locations*. The Lancet Planetary Health, 2021. **5**(9): p. e579-e587.
7. Johnson, B.T., et al., *Evaluation of biomass burning aerosols in the HadGEM3 climate model with observations from the SAMBBA field campaign*. Atmospheric Chemistry and Physics, 2016. **16**(22): p. 14657-14685.
8. Kollanus, V., et al., *Effects of long-range transported air pollution from vegetation fires on daily mortality and hospital admissions in the Helsinki metropolitan area, Finland*. Environ Res, 2016. **151**: p. 351-358.
9. Lima, C.H.R., A. AghaKouchak, and J.T. Randerson, *Unraveling the Role of Temperature and Rainfall on Active Fires in the Brazilian Amazon Using a Nonlinear Poisson Model*. Journal of Geophysical Research: Biogeosciences, 2018. **123**(1): p. 117-128.
10. Reddington, C.L., et al., *Air quality and human health improvements from reductions in deforestation-related fire in Brazil*. Nature Geoscience, 2015. **8**(10): p. 768-771.
11. Xu, R., et al., *Wildfires, Global Climate Change, and Human Health*. New England Journal of Medicine, 2020.
12. Zhao, Q., et al., *The association between heatwaves and risk of hospitalization in Brazil: A nationwide time series study between 2000 and 2015*. PLoS Med, 2019. **16**(2): p. e1002753.
13. Rodriguez-Cotto, R.I., et al., *Particle pollution in Rio de Janeiro, Brazil: increase and decrease of pro-inflammatory cytokines IL-6 and IL-8 in human lung cells*. Environ Pollut, 2014. **194**: p. 112-120.
14. UNEP, *Spreading like Wildfire: The Rising Threat of Extraordinary Landscape Fires*. 2022.

Reviewers' Comments:

Reviewer #1:

Remarks to the Author:

I think that the authors responded to most of the issues raised by the 3 reviewers. In my case, I am happy with the changes done in the manuscript in response to the issues raised. I also looked carefully at the answers from the other two reviewers, and the authors made a good job of improving the manuscript.

I am happy to recommend that NCOMMS accept the manuscript for publication.

Reviewer #2:

Remarks to the Author:

Thank you for considering reviewer comments and revising the manuscript. Your manuscript addresses an important question and has many of the right elements but still leaves some gaps that should be addressed.

The main limitation of this work is that PM due to wildfire smoke cannot be validated as such authors cannot make claims that per unit of PM they are more toxic than the other PM even if that may be true. The consequence can be illustrated by a following example. Suppose that the wildfire smoke PM_{2.5} concentrations are overestimated by a factor of 2; 1% increase in risk for 10 µg/m³ would turn out to be 0.5% increase in risk per 10 µg/m³. Now suppose they are overestimated by the factor of 10, the risk would go down to 0.05%. Because we don't know how far the estimate is from the truth, the authors cannot make statements comparing the risk from one source to comparing the risk from another source.

It would be ok to compare the attributable number and fraction, however from what I can discern attributable number is calculated at the fixed exposure level thus not reflecting the burden at the observed levels of pollution. As such the attributable numbers really reflect the population size.

Second limitation is the detail of the epidemiologic model. Since the first submission, the level of detail has dramatically improved but not without adding additional concerns. Where is the intercept in the model? How were the cumulative relative risks calculated, how were the pooled estimates calculated across multiple lags? How was RR calculated?

Third limitation is with the interpretation of wildfire PM. When wildfire smoke intrudes to any environment it is mixed with the background PM and with the anthropogenic sources of PM. The health risk between three sources cannot be discerned in the observational study. We simply breathe in the mixture.

Other questions:

Please explain the high PM in the northwest part of the state. How may the high levels as well as the daily variation in the non-wildfire PM impact the results.

Line 286 There are several other references you may want to use. One I am thinking about is Deflorio – Barker EHP circa 2019

Line 290 Don't start a sentence with "And"

Line 295. The reference #30 is a VERY good reference but somewhat dated on the issue many other studies have been published since and the new consensus seems to have been reached. There have also been some reviews published although they may be dated as well Adetona et al. 2016; Black et al. 2017a; Liu et al. 2015a; Reid et al. 2016a; Youssouf et al. but they may provide useful references.

Line 320 Please explain the biological mechanisms by which the following statement can be true. Wildfire-related air pollution could have lasting consequences and longer lag time than urban source PM_{2.5}.¹⁰

Seems like an overly sweeping statement but I may not be familiar with studies that could provide such evidence.

The whole paragraph on the strengths of the study is haphazardly put together and needs to be

reevaluated.

Please explain "To keep track" what are you referring to?

Line 323 "including wearing facemasks"- wearing facemasks is generally not recommended anymore. During covid we learned a lot about the filtering efficiencies of mask and know quite well that are not protective from the exposure to volatile compounds. I indeed smoke was worse than other sources in its toxicity it is generally thought if is because of the VOCs. Therefore masks would do little to protect from the added risk.

Please explain to the readers. What does staying indoors in the Amazon look like? Are there air conditioners? I know that towns like Manaus are big and developed but it is not clear that indoor temperatures are regularly controlled with ACs in a wider population.

Line 326 Authors do not provide a link between change in temperature and the risk of fire in the Brazil or stay closer to the evidence supported

"Immediate climate actions are required to limit the global mean temperature increase to 2.0°C or 1.5°C above preindustrial levels, avoiding 60% or 80%, respectively, of the increase in wildfire exposure.³⁵

"

Line 343 Please clarify how does the ozone amplify wildfire smoke health effects. Usually we think of ozone not being present due to impermeability of sun through the plumes. Sun is necessary to form ozone.

346 Please clarify what does this mean: Individual exposure representing by region- average tends to be independent of the true exposure level, and to be random.

Conclusion – needs to be rewritten. None of the statements are completely wrong but they just don't have a logical flow.

Reviewer #3:

Remarks to the Author:

I would like to thank the authors for their diligent and thorough revisions. All questions raised by me on the original manuscript have been responded to and I am satisfied that the revised manuscript can be published in its current form.

RESPONSE TO REVIEWER COMMENTS

Reviewer #1:

I think that the authors responded to most of the issues raised by the 3 reviewers. In my case, I am happy with the changes done in the manuscript in response to the issues raised. I also looked carefully at the answers from the other two reviewers, and the authors made a good job of improving the manuscript.

I am happy to recommend that NCOMMS accept the manuscript for publication.

[Response] We thank the reviewer for your time on reviewing this paper.

Reviewer #2:

Thank you for considering reviewer comments and revising the manuscript. Your manuscript addresses an important question and has many of the right elements but still leaves some gaps that should be addressed.

[Response] We thank the reviewer for the positive feedback on our study. We have addressed the remaining concerns point-by-point.

The main limitation of this work is that PM due to wildfire smoke cannot be validated as such authors cannot make claims that per unit of PM they are more toxic than the other PM even if that may be true. The consequence can be illustrated by a following example. Suppose that the wildfire smoke PM_{2.5} concentrations are overestimated by a factor of 2; 1% increase in risk for 10 ug/m³ would turn out to be 0.5% increase in risk per 10 ug/m³. Now suppose they are overestimated by the factor of 10, the risk would go down to 0.05%. Because we don't know how far the estimate is from the truth, the authors cannot make statements comparing the risk from one source to comparing the risk from another source. It would be ok to compare the attributable number and fraction, however from what I can discern attributable number is calculated at the fixed exposure level thus not reflecting the burden at the observed levels of pollution. As such the attributable numbers really reflect the population size.

[Response] We address each concern in turn.

(1) The effect estimates were consistent in magnitude with previous studies.

Currently, there have been many studies using the GEOS-Chem model to quantify the contributions of wildfire to regional and global PM_{2.5}¹⁻³, and they all showed that the GEOS-Chem model had good capability to isolate the fire-induced PM_{2.5}. In this study, we used the same model to simulate the PM_{2.5} concentration for the whole study period, not restricted to wildfire episodes. All-source PM_{2.5} (or we say total PM_{2.5}) from our model was validated and calibrated against global wide ground-level monitoring PM_{2.5} with an R² of 0.865. Then, wildfire-related PM_{2.5} was derived by multiplying all-source PM_{2.5} and a ratio (ranging from 0 to 1) of wildfire-related to all-source PM_{2.5}. We then conducted a time-series model to evaluate the association between wildfire-related PM_{2.5} and mortality.

With this well-developed time-series model, we estimated that each 10 µg/m³ increase in daily wildfire-related PM_{2.5} was associated with 3.1% (95% confidence interval [CI]: 2.4 to 3.9%) increase in all-cause mortality, 2.6% (95%CI: 1.5 to 3.8%) increase in cardiovascular mortality, and 7.7% (95%CI: 5.9 to 9.5) increase in respiratory mortality over 0–14 days. The RRs are consistent in magnitude with previous studies. Karanasiou et al. reviewed the health effects of biomass burning and summarized findings of 81 papers⁴. Their meta-analytic estimate was 1.92% (95%CI: 1.19 to 5.03) increased risk of all-cause mortality per 10 µg/m³ increase in PM_{2.5}. Each 10 µg/m³ increase in PM_{2.5} concentrations was associated with a 4.45% (95%CI: 0.96, 7.95) increased risk of cardiovascular mortality. The pooled effect estimate was 4.10% (95%CI: 2.86, 5.34) increased risk of total respiratory admissions/emergency visits per 10 µg/m³ increase in PM_{2.5}.

(2) Calculation of the attributable number and fraction.

For each immediate region i , we calculated the burden of mortality attributable to wildfire-related $PM_{2.5}$ as the attributable number of cause-specific deaths (AD) using previously published methods⁵

$$AD_{ti} = D_{ti} \times (RR_{ti} - 1) / RR_{ti}$$
$$RR_{ti} = \exp(\beta_i \times \Delta x_{ti})$$

where D_{ti} is the average daily death counts from day t to day $t+14$ in immediate region i ; RR_{ti} is the cumulative relative risk in immediate region i on day t associated with an increase in concentration of wildfire-related $PM_{2.5}$ from the above analyses (β_i); Δx_{ti} is the wildfire-related $PM_{2.5}$ concentration in immediate region i on day t . The 95%CI of AD was calculated by replacing the RR with its 95%CI bounds.

Following the above equations, attributable number was calculated at the daily level (daily exposure and daily death), thus reflecting the estimated mortality burden at the observed levels of pollution. Please also see our reply to comment#2 and Figure R1 for details.

For the comparison of the findings, we have stated in Discussion (Lines 309-319, clean version), “the estimated attributed deaths of our study are consistent in magnitude with those of previous investigations for South America and Brazil, despite different effect estimates and exposure periods, with strong evidence of acute adverse health outcomes due to exposure to wildfire-related $PM_{2.5}$. We estimated that 130,273 deaths could be attributable to wildfire-related $PM_{2.5}$ exposure from 2000 to 2016, equivalent to 7,663 deaths annually. Johnston et al estimated preventing fires would avoid 10,000 premature deaths annually between 1997 to 2006 in South America.⁶ Reddington et al estimated prevention of vegetation fires would avert about 7,000 to 9,700 premature deaths annually across South America and 4,200 to 5,200 in Brazil between 2002 and 2011⁷. A recent study estimated that vegetation fires in the Amazon basin in 2012, a year with emissions similar to the 11-year average (2008 to 2018), were linked to approximately 9,770 premature deaths in Brazil.⁸”

(3) We could not draw a conclusion of higher toxicity of wildfire-related $PM_{2.5}$ than the other PMs based on the findings.

A greater impact of wildfire-related $PM_{2.5}$ on mortality risks relative to ambient level cannot be concluded from our findings. However, stronger effects of wildfire-specific $PM_{2.5}$ than $PM_{2.5}$ from other sources has been observed in a recent epidemiological study⁹. For example, Aguilera et al. isolated wildfire-specific $PM_{2.5}$ using a series of statistical approaches and exposure definitions, and found higher increases in respiratory hospitalizations with increase in wildfire-specific $PM_{2.5}$ compared to the associations with non-wildfire $PM_{2.5}$ ⁹. A comparable study¹⁰ examining the elderly population in counties across the Western US found a 7.2% increase in the risk of respiratory admissions during smoke wave days with higher wildfire-specific $PM_{2.5}$ compared to matched nonsmoker days. Toxicological studies have shown differences in the toxicological characteristics of aerosols from different sources^{11 12}, and biomass particles exhibits greater toxicity in comparison with those produced by fossil fuels.

Second limitation is the detail of the epidemiologic model. Since the first submission, the level of detail has dramatically improved but not without adding additional concerns. Where is the intercept in the model? How were the cumulative relative risks calculated, how were the pooled estimates calculated across multiple lags? How was RR calculated?

[Response] We applied a distributed lag model which has been widely used by previous studies for time-series data¹³⁻¹⁵.

(1) Model building

We have modified the epidemiological model. Specifically, a quasi-Poisson regression with constrained lag model was applied for the time-series data for each immediate region:

$$Y_{ti} \sim \text{poisson}(\mu_{ti})$$
$$\text{Log}(\mu_{ti}) = \alpha + \text{cb}(PM_{2.5t}, df = 4) + \text{ns}(Temp_t, df = 3) + \text{ns}(RH_t, df = 3) + \text{ns}(Time_t, df = 7 \times n) + DOW_t + Holidays_t \quad (1)$$

Where Y_{ti} is the daily death count in immediate region i on day t ; α is the intercept. A cross-basis function was defined using a linear function for the space of $PM_{2.5}$, and a natural cubic spline for the space of 14 lag days with 4 degrees of freedom (df). We initially compared the lagged patterns of wildfire-related $PM_{2.5}$ on mortality during lags with the maximum of 7, 10 or 14 lag days (Figure S1). We found that a maximum of 14 lag days could fully capture the lagged effects. The model adjusted for the 21-day moving averages of daily mean temperature with a 3 df natural cubic spline and a 7-day moving average of daily mean relative humidity with 3 df natural cubic spline. Day of week and public holidays presented as categorical variables were also controlled in the models. Seasonality and long-term trends were controlled using a natural cubic spline of time with 7 degrees of freedom per year, a common choice for time-series studies.¹⁶

In the second stage, we pooled the immediate region-specific estimates at a national level by meta-analysis. The lagged effects and cumulative effects of wildfire-related $PM_{2.5}$ on mortality were described as the relative risk (RR) of mortality and corresponding 95% confidence intervals (CIs) per 10 $\mu\text{g}/\text{m}^3$ increase in $PM_{2.5}$ concentrations.

Under this framework (Figure R1), the exponentiated regression coefficient $RR = \exp(\beta * \Delta x)$ is the relative risk. β is the regression coefficient of wildfire-related $PM_{2.5}$ via the quasi-Poisson model. In the main text, we reported the effect estimates as relative risk with 95% CI of mortality associated with a 10 $\mu\text{g}/\text{m}^3$ increase in $PM_{2.5}$, i.e., $RR = \exp(\beta * 10)$.

Figure R1. The framework of statistical analysis.

(2) Delayed effects

We fitted the distributed lag linear model with the `dlnm` R package¹⁷.

In the presence of delayed effects, a cross-basis was applied. A cross-basis is a bi-dimensional space of functions describing simultaneously the shape of the relationship along exposure and its distributed lag effects. Choosing a cross-basis amounted to choosing two sets of basis functions, which were combined to generate the cross-basis functions. In this study, we specified the space of $\text{PM}_{2.5}$ as a linear function since our initial analysis (Figure S4), much like previous studies, showed that the PM –mortality association was linear, and defined a natural cubic spline for the space of 14 lag days with 4 degrees of freedom (df). This allows the effect of a single exposure event to be distributed over a specific period of time (the maximum lag period), using a spline function to explain the contributions at different lags. Finally, an estimate of the cumulative effect (or overall effect) could be computed by summing all the contributions at different lags.

The mathematical methods for cross-basis function and DLNM have been explained in detail by the developers¹⁷⁻¹⁹.

Third limitation is with the interpretation of wildfire PM . When wildfire smoke intrudes to any environment it is mixed with the background PM and with the anthropogenic sources of PM . The health risk between three sources cannot be discerned in the observational study. We simply breathe in the mixture.

[Response] Thanks for the comment. The population does breathe in the mixture, and wildfires can enhance the levels of such mixed air pollution. From our previous work on global fire air pollution simulation with GEOS-Chem model, we identified wildfire-related $\text{PM}_{2.5}$ from the background source. With an observational study applying a nationwide dataset and time-series model, we are able to examine the associations between exposure to wildfire-related $\text{PM}_{2.5}$ and cause-specific mortality. That is, this study aimed to provide epidemiological evidence on whether and to what extent wildfire-related $\text{PM}_{2.5}$ has health effects on human health. With the results, we observed statistically significant associations. Though we could not confirm greater toxicity of wildfire-related $\text{PM}_{2.5}$ with

this study, it has been of great interest to examine the toxicity of PMs from different sources. It is of great importance for air pollution control, as well as the reduction of PM-induced health burdens.

Other questions:

Please explain the high PM in the northwest part of the state. How may the high levels as well as the daily variation in the non-wildfire PM impact the results.

[Response] As we have replied to Review#3's comment#1 in the first-round revision, we added Figure R4 to show the spatial distribution of PM. Specifically, we extracted the gridded fractional source contributing to PM_{2.5} in Brazil from the 2021 Global Burden of Disease-Major Air Pollution Sources Study-Global (GBD_MAPS)²⁰ and added a map panel (Figure R4) to show the annual concentration of source-specific PM_{2.5}. We also added a map panel in Figure 1 to show the spatial distribution of wildfire-related PM_{2.5} and total PM_{2.5} during the study period. The spatial patterns of estimated wildfire-related PM_{2.5} by our study (Figure 1, panel A) was in line with the distribution of PM_{2.5} from open fires estimated by GBD_MAPS with higher concentrations in central west regions. Relative high concentration of total PM_{2.5} in north region (Figure 1, panel B) could be explained by PM_{2.5} from windblown dust and all remaining sources ("Other" in Figure R4). "Other" sources are primarily from non-combustion or uncategorized combustion sources (agriculture, solvents, biogenic secondary organic aerosol, waste incineration, etc.). We think this may explain the high concentration of total PM_{2.5} in these regions - as part of the Amazon rainforest- where dead plants and animal matter accumulate continuously.

As we stated above, we identified wildfire-related PM_{2.5} from the background sources. The associations between wildfire-related PM_{2.5} and mortality were examined independently. The spatiotemporal variation in the non-wildfire PM should not affect the associations.

Figure R4. Annual concentration of source-specific PM_{2.5} in Brazil for the year 2017. (Data source: 2021 Global Burden of Disease-Major Air Pollution Sources Study-Global²⁰) Adopted from Pei et al²¹.

Line 286 There are several other references you may want to use. One I am thinking about is Deflorio – Barker EHP circa 2019

[Response] Thanks for the suggestion. We have added this reference in the revision.

Line 290 Don't start a sentence with "And"

[Response] We rewrote the sentence as "The sex differences for all-cause and respiratory mortalities remain significant in ≥ 60 years age subgroups (Figure S4)."

Line 295. The reference #30 is a VERY good reference but somewhat dated on the issue many other studies have been published since and the new consensus seems to have been reached. There have also

been some reviews published although they may be dated as well Adetona et al. 2016; Black et al. 2017a; Liu et al. 2015a; Reid et al. 2016a; Youssouf et al. but they may provide useful references.

[Response] Thanks for the suggested review articles. We have read all of them. In Line 295, we discussed that in line with our results, there are also studies reporting higher risks in females than males for cardiovascular diseases in relation to wildfire smoke,^{22 23} but inconsistency remains. We have cited a review study to highlight the inconsistency. We replaced reference#30 with the review by Black et al. published in 2017.

Line 320 Please explain the biological mechanisms by which the following statement can be true. Wildfire-related air pollution could have lasting consequences and longer lag time than urban source PM_{2.5}.¹⁰ Seems like an overly sweeping statement but I may not be familiar with studies that could provide such evidence.

[Response] “Wildfire-related air pollution could have longer lag time than urban source PM_{2.5},²⁴” is epidemiological evidence confirmed by our previous work²⁴ in which stronger effects on mortality (higher RRs) and longer lag effects (Lag 0-3 days) were found by comparison with a previous study by Liu et al.²⁵ These two studies were conducted with the same multi-country-multi-city dataset. Liu et al. evaluated the associations of total PM_{2.5} (mainly from urban background) and daily mortality and observed that the effects were significant on lag0-2 days. As reviewer commented, we cannot directly compare the risks (effect sizes) of two studies using air pollution data from different sources/models. We admit the statement might be less reliable, and have rewritten the lines.

To our knowledge, there is no direct biological evidence on comparing the lasting consequences of PM_{2.5} from different sources. However long-lasting of wildfire smoke have been reported. Smoke from fires can linger in the atmosphere for days, weeks or even months²⁶.

Though the differential toxicity of wildfire PM_{2.5} as compared to other ambient sources of PM_{2.5} is not well understood, toxicological studies have shown differences in the composition and effects of wildfire PM_{2.5} compared to ambient background sources²⁷. In epidemiological studies, greater impact of wildfire-specific PM_{2.5} than PM_{2.5} from other sources has been observed⁹. In this recent study, Aguilera et al. isolated wildfire-specific PM_{2.5} using a series of statistical approaches and exposure definitions, and found higher increases in respiratory hospitalizations with increase in wildfire-specific PM_{2.5} compared to the associations with non-wildfire PM_{2.5}. A comparable study¹⁰ examining the elderly population in counties across the Western US found a 7.2% increase in the risk of respiratory admissions during smoke wave days with higher wildfire-specific PM_{2.5} compared to matched nonsmoker days.

We have removed these lines in Discussion.

The whole paragraph on the strengths of the study is haphazardly put together and needs to be reevaluated.

[Response] “This study provides robust epidemiological evidence for mortality risk attributable to short-term exposure to wildfire-related PM_{2.5}, based on a large nationwide dataset in Brazil. The findings were not only representative of the general Brazilian population, but could provide information for assessing the mortality risks and burdens from acute wildfire-related PM_{2.5}”

exposure in other countries and regions with the similar population demographics, healthcare facilities, and socioeconomic status. Consistent evidence suggested associations between wildfire smoke exposure and respiratory diseases; however, evidence on circulatory health was limited. Ours is the first and largest research study to characterize the relationship between exposure to wildfire-related PM_{2.5} and mortality for cardiovascular diseases in Brazil. We also observed geographic and demographic variations in these associations.”

Please explain “To keep track” what are you referring to?

[Response] We suggest to keep track of air quality during the fire seasons.

“We suggest public agencies that are responsible for releasing advice regarding health protection against wildfire smoke educating residents keep track of air quality during fire season.²⁸ It is vital for residents living in areas potentially affected by wildfires to adjust their activities, and gather emergency supplies (e.g., food, water, first aid medication) before a fire occurs.”

Line 323 “including wearing facemasks”- wearing facemasks is generally not recommended anymore. During covid we learned a lot about the filtering efficiencies of mask and know quite well that are not protective from the exposure to volatile compounds. I indeed smoke was worse than other sources in its toxicity it is generally thought if is because of the VOCs. Therefore masks would do little to protect from the added risk.

[Response] Whilst mask is not protective from the exposure to volatile compounds, P2 or N95 masks do filter out most but not all fine particles. We rewrote the sentence.

“Personal protections are recommended, including wearing facemasks, avoiding heavy and prolonged physical activity, staying indoors and keeping the windows closed, using air purifiers, especially for sensitive populations.”

Please explain to the readers. What does staying indoors in the Amazon look like? Are there air conditioners? I know that towns like Manaus are big and developed but it is not clear that indoor temperatures are regularly controlled with ACs in a wider population.

[Response] Staying indoors was a general suggestion. Wildfire-related air pollution can travel widely, threatening people living across regions in Brazil. For many residents in the Amazon, a staying indoors strategy might be less effective or even impractical. We suggested further assistance from local governments.

Line 326 Authors do not provide a link between change in temperature and the risk of fire in the Brazil or stay closer to the evidence supported

“Immediate climate actions are required to limit the global mean temperature increase to 2.0°C or 1.5°C above preindustrial levels, avoiding 60% or 80%, respectively, of the increase in wildfire exposure.”

[Response] As stated in the first sentence of this paragraph, “These findings could aid in raising awareness of the wildfire smoke crisis, and responses to protect health from wildfire-related air pollution.” However, risk warming and personal protections during a wildfire event are insufficient. The increased intensity and frequency of fires in Brazil could be partially attributable to climate

change and global warming trend – dry weather, wind and heat. Actions to cut carbon emissions would limit global temperature increases and mitigate wildfire risks, ultimately, benefit the population. We deleted this sentence and rewrote the whole paragraph as:

“These findings could aid in raising awareness of the wildfire smoke crisis, and responses to protect health from wildfire-related air pollution. We suggest public agencies that are responsible for releasing advice regarding health protection from wildfire smoke educate residents to keep track of air quality during fire season.²⁸ It is vital for residents living in areas potentially affected by wildfires to adjust their activities, and gather emergency supplies (e.g., food, water, first aid, medication) before a fire occurs. Personal protections are recommended, including wearing facemasks, avoiding heavy and prolonged outdoor activities, staying indoors and keeping the windows closed, using air purifiers, especially for sensitive populations. For residents in the Amazon, a staying indoors strategy might be less effective or even impractical. We suggest further assistance from local governments.”

Line 343 Please clarify how does the ozone amplify wildfire smoke health effects. Usually we think of ozone not being present due to impermeability of sun through the plumes. Sun is necessary to form ozone.

[Response] Wildfire activity is an important source of tropospheric ozone. Wildfires can contribute to tropospheric ozone by releasing large amount of ozone precursors, i.e., nitrogen oxides and volatile organic compounds, which form ozone via reacting in the presence of sunlight^{29 30}. Kalashnikov et al. identified significant increases in the frequency, spatial extent, and temporal persistence of extreme PM_{2.5}/ozone co-occurrences, and linked the spatial extent of co-occurrence to the extent of extreme heat and wildfires. The co-occurrences increased population exposure to multiple harmful air pollutants. A growing body of studies has examined the adverse health effects of wildfire-related ozone³¹⁻³³. In one study, Reid et al.³² found significant relationships between ozone and PM_{2.5} and respiratory hospitalizations and emergency department visits in mutually-adjusted models, however, inconsistency remained. The authors recommend future studies of the health impacts of wildfire smoke incorporate exposures to ozone as well as PMs to get a better perspective on the complete health impacts of wildfires. A study in Western Amazon (Brazilian Amazon) found that high dose of ozone posed a toxicological risk on schoolchildren located in an area of biomass burning activities³³.

346 Please clarify what does this mean: Individual exposure representing by region- average tends to be independent of the true exposure level, and to be random.

[Response] We discussed the uncertainties in exposure assessment due to using averaged air pollution at region level to represent individual exposure. However, this error is mainly of Berkson type (i.e., statistically uncorrelated with the observed variable).³⁴⁻³⁶ This does not substantially bias effect estimates for the associations between exposure and response, but dose lose precision, i.e., make the confidence intervals wider.

Conclusion – needs to be rewritten. None of the statements are completely wrong but they just don't have a logical flow.

[Response] Thanks for the suggestion. The UNEP report referenced in Discussion was suggested by reviewer#3 in the first round of revision. A couple of sentences reflecting on what the findings of

this manuscript would mean under conditions of increasing wildfire risks due to climate change in the region may be worth adding, to further increase the potential impact of our paper.

“Our findings suggested that short-term exposure to wildfire-related PM_{2.5} is associate with increase in mortality risks of all-cause, cardiovascular, and respiratory, even for populations living further away from the burnt areas. We also observed stronger associations among females and older adults aged ≥ 60 years. These findings have important implications for adaptation strategies and emergency planning to better mitigate wildfire-related health risks under conditions of increasing wildfire risks in Brazil³⁷. For example, public health professionals should educate residents raising awareness of wildfire smoke crisis and also guide prompt public responses and take actions to reduce exposure, especially for sensitive populations.”

Reviewer #3:

I would like to thank the authors for their diligent and thorough revisions. All questions raised by me on the original manuscript have been responded to and I am satisfied that the revised manuscript can be published in its current form.

[Response] We thank the reviewer for your time on reviewing this paper.

References

1. Zhu J, Yue X, Che H, et al. Contribution of Fire Emissions to PM_{2.5} and Its Transport Mechanism Over the Yungui Plateau, China During 2015–2019. *Journal of Geophysical Research: Atmospheres* 2022;127(12) doi: 10.1029/2022jd036734
2. Carter TS, Heald CL, Cappa CD, et al. Investigating Carbonaceous Aerosol and Its Absorption Properties From Fires in the Western United States (WE-CAN) and Southern Africa (ORACLES and CLARIFY). *Journal of Geophysical Research: Atmospheres* 2021;126(15) doi: 10.1029/2021jd034984
3. O'Dell K, Ford B, Fischer EV, et al. Contribution of Wildland-Fire Smoke to US PM_{2.5} and Its Influence on Recent Trends. *Environ Sci Technol* 2019;53(4):1797-804. doi: 10.1021/acs.est.8b05430 [published Online First: 2019/01/27]
4. Karanasiou A, Alastuey A, Amato F, et al. Short-term health effects from outdoor exposure to biomass burning emissions: A review. *The Science of the total environment* 2021;781:146739. doi: 10.1016/j.scitotenv.2021.146739 [published Online First: 2021/04/03]
5. Xu R, Zhao Q, Coelho M, et al. The association between heat exposure and hospitalization for undernutrition in Brazil during 2000-2015: A nationwide case-crossover study. *PLoS medicine* 2019;16(10):e1002950. doi: 10.1371/journal.pmed.1002950 [published Online First: 2019/10/30]
6. Johnston FH, Henderson SB, Chen Y, et al. Estimated global mortality attributable to smoke from landscape fires. *Environmental health perspectives* 2012;120(5):695-701. doi: 10.1289/ehp.1104422 [published Online First: 2012/03/30]
7. Reddington CL, Butt EW, Ridley DA, et al. Air quality and human health improvements from reductions in deforestation-related fire in Brazil. *Nature Geoscience* 2015;8(10):768-71. doi: 10.1038/ngeo2535
8. Butt EW, Conibear L, Reddington CL, et al. Large air quality and human health impacts due to Amazon forest and vegetation fires. *Environmental Research Communications* 2020;2(9) doi: 10.1088/2515-7620/abb0db
9. Aguilera R, Corringham T, Gershunov A, et al. Wildfire smoke impacts respiratory health more than fine particles from other sources: observational evidence from Southern California. *Nat Commun* 2021;12(1):1493. doi: 10.1038/s41467-021-21708-0 [published Online First: 2021/03/07]
10. Liu JC, Wilson A, Mickley LJ, et al. Wildfire-specific Fine Particulate Matter and Risk of Hospital Admissions in Urban and Rural Counties. *Epidemiology* 2017;28(1):77-85. doi: 10.1097/EDE.0000000000000556 [published Online First: 2016/09/21]
11. Mazzoli-Rocha F, Carvalho GM, Lanzetti M, et al. Respiratory toxicity of repeated exposure to particles produced by traffic and sugar cane burning. *Respir Physiol Neurobiol* 2014;191:106-13. doi: 10.1016/j.resp.2013.11.004 [published Online First: 2013/11/28]
12. Mazzoli-Rocha F, Oliveira VR, Barcellos BC, et al. Time-dependency of mice lung recovery after a 4-week exposure to traffic or biomass air pollutants. *Respir Physiol Neurobiol* 2016;230:16-21. doi: 10.1016/j.resp.2016.05.003 [published Online First: 2016/05/18]
13. Zhao Q, Zhang Y, Zhang W, et al. Ambient temperature and emergency department visits: Time-series analysis in 12 Chinese cities. *Environmental pollution* 2017;224:310-16. doi: 10.1016/j.envpol.2017.02.010 [published Online First: 2017/02/23]
14. Zhao Q, Li S, Coelho M, et al. The association between heatwaves and risk of hospitalization in Brazil: A nationwide time series study between 2000 and 2015. *PLoS medicine* 2019;16(2):e1002753. doi: 10.1371/journal.pmed.1002753
15. Hu K, Guo Y, Hu D, et al. Mortality burden attributable to PM₁ in Zhejiang province, China. *Environment international* 2018;121(Pt 1):515-22. doi: 10.1016/j.envint.2018.09.033

16. Bhaskaran K, Gasparrini A, Hajat S, et al. Time series regression studies in environmental epidemiology. *International journal of epidemiology* 2013;42(4):1187-95. doi: 10.1093/ije/dyt092
17. Gasparrini A. Distributed Lag Linear and Non-Linear Models in R: The Package dlnm. *J Stat Softw* 2011;43(8):1-20. [published Online First: 2011/10/18]
18. Gasparrini A, Armstrong B, Kenward MG. Distributed lag non-linear models. *Statistics in medicine* 2010;29(21):2224-34. doi: 10.1002/sim.3940
19. Gasparrini A, Armstrong B. Reducing and meta-analysing estimates from distributed lag non-linear models. *BMC Medical Research Methodology* 2013;13(1):1. doi: 10.1186/1471-2288-13-1
20. McDuffie EE, Martin RV, Spadaro JV, et al. Source sector and fuel contributions to ambient PM2.5 and attributable mortality across multiple spatial scales. *Nat Commun* 2021;12(1):3594. doi: 10.1038/s41467-021-23853-y [published Online First: 2021/06/16]
21. Yu P, Xu R, Li S, et al. Exposure to wildfire-related PM2.5 and site-specific cancer mortality in Brazil from 2010 to 2016: A retrospective study. *PLoS medicine* 2022;19(9):e1004103. doi: 10.1371/journal.pmed.1004103 [published Online First: 2022/09/20]
22. Tinling MA, West JJ, Cascio WE, et al. Repeating cardiopulmonary health effects in rural North Carolina population during a second large peat wildfire. *Environmental health : a global access science source* 2016;15:12. doi: 10.1186/s12940-016-0093-4 [published Online First: 2016/01/29]
23. Jones CG, Rappold AG, Vargo J, et al. Out-of-Hospital Cardiac Arrests and Wildfire-Related Particulate Matter During 2015-2017 California Wildfires. *J Am Heart Assoc* 2020;9(8):e014125. doi: 10.1161/JAHA.119.014125 [published Online First: 2020/04/16]
24. Chen G, Guo Y, Yue X, et al. Mortality risk attributable to wildfire-related PM2.5 pollution: a global time series study in 749 locations. *The Lancet Planetary Health* 2021;5(9):e579-e87. doi: 10.1016/s2542-5196(21)00200-x
25. Liu C, Chen R, Sera F, et al. Ambient Particulate Air Pollution and Daily Mortality in 652 Cities. *New England Journal of Medicine* 2019;381(8):705-15. doi: 10.1056/NEJMoa1817364
26. Yu P, Toon OB, Bardeen CG, et al. Black carbon lofts wildfire smoke high into the stratosphere to form a persistent plume. *Science* 2019;365(6453):587-90. doi: doi:10.1126/science.aax1748
27. Kim YH, Warren SH, Krantz QT, et al. Mutagenicity and Lung Toxicity of Smoldering vs. Flaming Emissions from Various Biomass Fuels: Implications for Health Effects from Wildland Fires. *Environmental health perspectives* 2018;126(1):017011. doi: 10.1289/EHP2200 [published Online First: 2018/01/27]
28. Xu R, Yu P, Abramson MJ, et al. Wildfires, Global Climate Change, and Human Health. *New England Journal of Medicine* 2020 doi: 10.1056/NEJMSr2028985
29. Jaffe DA, Wigder NL. Ozone production from wildfires: A critical review. *Atmospheric Environment* 2012;51:1-10. doi: 10.1016/j.atmosenv.2011.11.063
30. Watson GL, Telesca D, Reid CE, et al. Machine learning models accurately predict ozone exposure during wildfire events. *Environmental pollution* 2019;254(Pt A):112792. doi: 10.1016/j.envpol.2019.06.088 [published Online First: 2019/08/20]
31. Meo SA, Abukhalaf AA, Alomar AA, et al. Effect of environmental pollutants PM-2.5, carbon monoxide, and ozone on the incidence and mortality of SARS-COV-2 infection in ten wildfire affected counties in California. *The Science of the total environment* 2021;757:143948. doi: 10.1016/j.scitotenv.2020.143948 [published Online First: 2020/12/16]
32. Reid CE, Considine EM, Watson GL, et al. Associations between respiratory health and ozone and fine particulate matter during a wildfire event. *Environment international* 2019;129:291-98. doi: 10.1016/j.envint.2019.04.033 [published Online First: 2019/05/31]

33. Silva PR, Ignotti E, Oliveira BF, et al. High risk of respiratory diseases in children in the fire period in Western Amazon. *Rev Saude Publica* 2016;50 doi: 10.1590/S1518-8787.2016050005667 [published Online First: 2016/06/16]
34. Heid IM, Küchenhoff H, Miles J, et al. Two dimensions of measurement error: classical and Berkson error in residential radon exposure assessment. *Journal of exposure analysis and environmental epidemiology* 2004;14(5):365-77. doi: 10.1038/sj.jea.7500332 [published Online First: 2004/09/14]
35. Armstrong BG. Effect of measurement error on epidemiological studies of environmental and occupational exposures. *Occupational and environmental medicine* 1998;55(10):651-6. doi: 10.1136/oem.55.10.651 [published Online First: 1999/02/04]
36. Szpiro AA, Sheppard L, Lumley T. Efficient measurement error correction with spatially misaligned data. *Biostatistics (Oxford, England)* 2011;12(4):610-23. doi: 10.1093/biostatistics/kxq083
37. UNEP. Spreading like Wildfire: The Rising Threat of Extraordinary Landscape Fires, 2022.

Reviewers' Comments:

Reviewer #2:

Remarks to the Author:

The authors improved the manuscript significantly.

RESPONSE TO REVIEWER COMMENTS

Reviewer #2 (Remarks to the Author):

The authors improved the manuscript significantly.

[Response] We thank the reviewer for your time on reviewing this paper.